# AN INFORMATION-THEORETIC APPROACH TO DIVERSITY EVALUATION OF PROMPT-BASED GENERATIVE MODELS

## ABSTRACT

Text-conditioned generation models are commonly evaluated based on the quality of the generated data and its alignment with the input text prompt. On the other hand, several applications of prompt-based generative models require sufficient diversity in the generated data to ensure the models' capability of generating image and video samples possessing a variety of features. However, the existing diversity metrics are designed for unconditional generative models, and thus cannot distinguish the diversity arising from variations in text prompts and that contributed by the generative model itself. In this work, our goal is to quantify the prompt-induced and model-induced diversity in samples generated by prompt-based models. Specifically, we propose the application of matrix-based information measures to address this task, decomposing the kernel-based entropy $H(X)$ of generated data $X$ into the sum of conditional entropy $H(X|T)$, given text variable $T$, and mutual information $I(X;T)$. We show that this information-theoretic approach decomposes the existing Vendi diversity score defined based on $H(X)$ into the product of the following two terms: 1) *Conditional-Vendi* score based on $H(X|T)$ to quantify the model-induced diversity, and 2) *Information-Vendi* score based on $I(X;T)$ to measure the statistical relevance between $X$ and prompt $T$. Our theoretical results provide an interpretation for this diversity quantification and show that the Conditional-Vendi score aggregates the Vendi scores within the modes of a mixture prompt distribution. We conduct several numerical experiments to show the correlation between the Conditional-Vendi score and the internal diversity of text-conditioned generative models.

## 1 INTRODUCTION

Prompt-based generative models, including text-to-image and text-to-video generation schemes, are widely used in various artificial intelligence (AI) applications. In prompt-based generative AI, the sample creation process begins with a text input and produces a random output aligned with that text. The conditional nature of this sample generation distinguishes prompt-based generative models from standard unconditional generative models where the objective is to produce samples distributed similarly to real data without any guiding input prompt. Since most evaluation metrics for generative models had been developed for unconditional models in the previous decade, the recent literature has sought to create scores tailored specifically for text-conditioned generative models.

The existing evaluation metrics for prompt-based generative models typically focus on fidelity and relevance in sample generation, i.e., they assess the visual quality of the produced samples and their alignment with the input prompt. Relevance is often measured by calculating a similarity score between a shared embedding of the text and image samples, e.g. in ClipScore (Hessel et al., 2021) which utilizes the CLIP embeddings of text and image data. Such shared embedding-based evaluation mechanisms have been further adapted to quantify the aesthetics, semantic consistency, and compositional accuracy of the generated data based on the input text prompt.

On the other hand, the diversity performance of prompt-based generative models has not been exclusively studied in the literature. The diversity scores proposed for unconditional generative models, such as Recall (Sajjadi et al., 2018; Kynkäänniemi et al., 2019), Coverage (Naeem et al., 2020),

**Prompt-induced Diversity** | **Model-induced Diversity**

A person eating lunch    Someone is drinking tea      A person is drinking tea    Someone is drinking tea

A man reading a newspaper    Somebody sitting at a cafe      A person drinks tea    Somebody drinks tea

Figure 1: Illustration of *prompt-induced diversity*, where the diversity of generated images follows the variety of prompts and has little variation in other details, vs. *model-induced diversity*, where the diversity of images for similar prompts is due to the generation model.

Vendi (Friedman & Dieng, 2023; Pasarkar & Dieng, 2024), and RKE (Jalali et al., 2023), are often applied to quantify the variety of generated samples. However, in text-based generative models, the generated samples are typically produced in response to different input prompts, where the variation in input texts can significantly contribute to the diversity of the generated image or video samples. Thus, the diversity of data produced by prompt-based models is influenced by two main factors: 1) the variety of input prompts, and 2) the internal diversity of the model introducing randomness into the output samples. Figure 1 illustrates examples of prompt-induced diversity, where the variety of generated images is mostly due to the different prompts and the generated data has little variety in other details (images of similar people with similar poses), and model-induced diversity where the details not specified by the prompts vary significantly between generated images. This decomposition of diversity in text-based generative models has not been studied in the existing literature on conditional generative models including text-based image and video generation.

In this work, we focus on quantifying the two diversity components mentioned for prompt-based generative models. To this end, we propose an information-theoretic decomposition of the diversity of the model's output data, $X$. The proposed decomposition is based on a classical identity in information theory, which shows that for variables $X$ and $T$, the Shannon entropy $H(X)$, representing the uncertainty of $X$, can be decomposed into two terms as follows:

$$H(X) \;=\; H\big(X|T\big) + I\big(X;T\big)$$

Here, $H(X|T)$ denotes the conditional entropy of data $X$ given the text variable $T$, which we interpret as the internal diversity of the text-based generative model not caused by variation in the input text $T$. Furthermore, the mutual information term $I(X;T)$ can be viewed as a measure of statistical relevance between the text $T$ and the generated data $X$, quantifying how much information the model's output conveys about the input text.

To mathematically define the entropy-based scores, we follow the kernel-matrix-based entropy definition, which has been applied by (Friedman & Dieng, 2023; Jalali et al., 2023; Pasarkar & Dieng, 2024) to unconditional generative models. These references apply the matrix-based entropy in quantum information theory, that is the entropy of the eigenvalues of the kernel matrix of generated data $X$, to measure the diversity of an unconditional model's generated data. To extend the framework to conditional prompt-based generative models, we utilize the definition of matrix-based conditional entropy proposed by Giraldo et al. (2014). This work provides a definition for the conditional entropy of two general positive semi-definite matrices, which we select to be the kernel matrices of generated data $X$ and text $T$. Following these definitions, our work extends the entropy-based approach in (Friedman & Dieng, 2023; Jalali et al., 2023) to conditional generative models. We de-

fine the *Conditional-Vendi* and *Information-Vendi* scores which decompose the unconditional Vendi score to model-induced and prompt-induced diversity measures.

To statistically interpret the defined scores, we derive the statistic estimated by the proposed scores from empirical generated samples. We show how the target statistic can be formulated in terms of the kernel covariance matrix of the Kronecker product of text $T$ and data $X$ vectors. Importantly, we prove a theorem to interpret and provide an operational meaning for the proposed entropy-based scores. Note that the conditional entropy measures in (Giraldo et al., 2014) do not follow standard information theory, and hence the defining equation $H(X,T) - H(T) = \mathbb{E}_{t \sim P_T}[H(X|T = t)]$, well-known for Shannon conditional entropy, does not hold for the conditional entropy measure in (Giraldo et al., 2014). To address this gap and provide a practical interpretation for the defined conditional entropy $H(X|T)$, we prove that under a mixture distribution $P_T$ for text data with a hidden group variable $G$, the defined conditional entropy indeed aggregates $H(X|G = g)$ given prompt type $g \sim P_G$. This result shows the Conditional-Vendi is an aggregation of Vendi scores.

Specifically, our theoretical analysis on the connection between the Conditional VENDI score and average of unconditional entropy scores indicate a spectral approach to interpret the diversity evaluation of the proposed score. Following the spectral identification of modes in the eigendecomposition of matrix-based entropy function, we show that the modes used by the Conditional-VENDI score follow from the eigenvectors of the kernel covariance matrix of the text prompt distributions. Therefore, we visualize the diversity contributed by the prompt-based generative model using the spectral clustering of text samples according to the kernel function used in the definition of the diversity score. After computing the eigendirections of the text kernel matrix, we analyze the Hadamard product of each eigendirection-based rank-1 matrix and the joint (prompt,data) kernel matrix. This analysis reveals the variant clusters of images generated for each group of input text samples.

We numerically evaluate the proposed diversity scores for standard text-to-image, text-to-video, and image-captioning generative models. In our experiments, we simulate text-based generative models for which the ground-truth rankings of internal diversity and relevance are known. Our experimental results validate the consistency of our proposed information-theoretic scores and the ground-truth ranking of the models. We further decompose the Conditional-Vendi score across different modes of input text data, evaluating the models' internal diversity across different types of input text. The following is a summary of the contributions of this work:

- Highlighting the diversity evaluation task in the context of conditional generative models,
- Proposing an information-theoretic framework for decomposing the diversity of generated data into prompt-induced and model-induces components to evaluate the internal diversity of prompt-based generative models
- Providing an operational meaning for the defined scores and interpreting them as the average of entropy scores over the modes of a multi-modal text distribution
- Presenting numerical results on the consistency between the conditional entropy score and the model-induced diversity of text-to-image and text-to-video generative models.

## 2 RELATED WORK

**Evaluation of deep generative models**: The existing metrics for the evaluation of generative models can be divided into reference-dependent and reference-free categories, as discussed in (Borji, 2022). As one type of reference-dependent metrics, a distance between generated and reference distributions is measured using metrics such as FID (Heusel et al., 2017) and KID (Bińkowski et al., 2018). Other reference-based metrics such as the Inception Score (Salimans et al., 2016), GAN-train/GAN-test (Shmelkov et al., 2018), Precision/Recall (Sajjadi et al., 2018; Kynkäänniemi et al., 2019), and Density/Coverage (Naeem et al., 2020) are defined to quantify the diversity and quality of generated data in comparison to the samples in the real dataset. In addition, assessing memorization and novelty has been studied in several references, including the authenticity score (Alaa et al., 2022) and Feature Likelihood Divergence (Jiralerspong et al., 2023) to assess generalizability, and the rarity score (Han et al., 2023), KEN (Zhang et al., 2024) proposed to assess novelty. Note that the memorization metrics are inherently reference-based. In contrast, reference-free evaluations measure diversity and quality based only based on the generated data. Specifically, the Vendi (Friedman & Dieng, 2023; Pasarkar & Dieng, 2024) and RKE scores (Jalali et al., 2023) fall into this category.

**Evaluation of conditional generative models**: The evaluation of prompt-based generative models, including text-to-image and text-to-video models, has been studied in several related works. Most of the existing evaluation metrics attempt to measure the correlation between the prompt and the output. A standard metric for measuring the alignment of prompt and image is CLIPScore (Hessel et al., 2021), which measures the cosine similarity of the paired data using CLIP embedding. Some other works introduce different benchmarks and sets of prompts to evaluate different aspects. HEIM (Lee et al., 2023) assesses twelve aspects of the sample generation, including text-image alignment, image quality and bias. As noted by Astolfi et al. (2024), standard metrics focusing on style, aesthetics, and image quality, may overlook the diversity of images given a particular prompt. In their work, they measure diversity separately for each prompt using a similarity function and average the scores for the prompts. Kannen et al. (2024) follow a similar approach with the Vendi score. We note that both these methods require generating multiple images per prompt with different seeds to measure the score. On the other hand, our proposed Conditional-Vendi score does not require multiple sample generations per prompt and instead analyzes the types of text prompt in the assessment. Our theoretical results interpret Conditional-Vendi as an aggregation of the scores over text types.

**Information measures for evaluating conditional generative models**: Kim et al. (2022) utilize the mutual information (MI) between continuous text and image variables, and propose the Mutual Information Divergence (MID) score. This work fits a multivariate Gaussian distribution to the text and image data and then estimates their mutual information to quantify a relevance metric for conditional generative models. We note that our proposed Information-Vendi score is based on the matrix-based entropy score by Giraldo et al. (2014) which is different from the MI between Gaussian vectors fitted to the text and image data used in the MID score. Different from MID, Information-Vendi relies on kernel similarity values to identify a cluster variable for MI calculation.

## 3 PRELIMINARIES

Throughout the work, we focus on a conditional generative model that produces a data vector $X \in \mathcal{X}$ given an input text prompt $T \in \mathcal{T}$ according to a conditional distribution $P_{X|T}$, i.e., for text prompt $T = t$ the model outputs a random sample following $P_{X|T=t}$. We consider $n$ sample pairs $(t_i, x_i) \sim P_T \times P_{X|T}$ where each text prompt $t_i$ is drawn independently from the distribution $P_T$ and then the generated sample $x_i$ is generated according to $P_{X|T=t_i}$. Our goal is to quantify the internal diversity of the prompt-based generative model, influencing the variety of data generated $x_1, \ldots, x_n$ independently of the diversity of input texts $t_1, \ldots, t_n$.

### 3.1 ENTROPY-BASED DIVERSITY SCORES FOR UNCONDITIONAL GENERATIVE MODELS

Consider generated samples $x_1, \ldots, x_n \in \mathcal{X}$ following the distribution $P_X$ of an unconditional generative model. For a kernel function $k : \mathcal{X} \times \mathcal{X} \to \mathbb{R}$, the kernel similarity matrix $K \in \mathbb{R}^{n \times n}$ is $K = \left[ k(x_i, x_j) \right]_{1 \leq i,j \leq n}$. Following the standard definition, a kernel function $k$ satisfies the positive semidefinite property (PSD), which means that the above kernel matrix will be PSD for any arbitrary selection of data points $x_1, \ldots, x_n \in \mathcal{X}$, i.e., all its eigenvalues are non-negative. A popular kernel function is the Gaussian (RBF) kernel, which for a bandwidth parameter $\sigma$ is defined as:

$$k(x, x') = \exp\left(\frac{-\left\|x - x'\right\|_2^2}{2\sigma^2}\right) \tag{1}$$

Assuming that a kernel function $k$ is normalized, i.e. $k(x, x) = 1$ for every $x \in \mathbb{R}^d$, then the non-negative eigenvalues $\lambda_1, \ldots, \lambda_n$ of $\frac{1}{n}K$ will add up to 1, implying that they represent a probability model. In the literature, Friedman & Dieng (2023); Jalali et al. (2023); Pasarkar & Dieng (2024) propose using the general order-$\alpha$ Renyi entropy of the probability model as the model's diversity score, defined as follows for $\frac{1}{n}K$:

$$H_\alpha(X) := H_\alpha\left(\frac{1}{n}K\right) = \frac{1}{1-\alpha} \log\left(\sum_{i=1}^{n} \lambda_i^\alpha\right) \tag{2}$$

In the special case of $\alpha = 1$, the above definition results in the Shannon-entropy of eigenvalues $H_1(\frac{1}{n}K) = \sum_{i=1}^{n} \lambda_i \log(1/\lambda_i)$. Also, we note that the Vendi and RKE scores defined by Friedman & Dieng (2023); Pasarkar & Dieng (2024) are the exponential of the defined entropy measure, where

$$\text{Vendi}_\alpha(x_1, \ldots, x_n) := \exp\left(H_\alpha\left(\frac{1}{n}K\right)\right) \tag{3}$$

To statistically interpret the entropy measures of the samples' kernel matrix, Bach (2022); Jalali et al. (2023) note that the normalized kernel matrix $\frac{1}{n}K$ shares the same non-zero eigenvalues with the empirical kernel covariance matrix $\widehat{C}_X$ defined as:

$$\widehat{C}_X = \frac{1}{n}\sum_{i=1}^{n} \phi(X_i)\phi(X_i)^{\top} \tag{4}$$

Here, $\phi : \mathcal{X} \to \mathbb{R}^d$ denotes the kernel feature map satisfying the relation $k(x, x') = \langle \phi(x), \phi(x') \rangle$ for every $x, x' \in \mathcal{X}$ where $\langle \cdot, \cdot \rangle$ is the standard inner-product in the $\mathbb{R}^d$ space. As a result, the entropy of the kernel matrix $K$'s eigenvalues equals the entropy of $\widehat{C}_X$. Note that $\widehat{C}_X$ is the empirical estimation of the underlying kernel covariance matrix $C_X = \mathbb{E}_{X \sim P_X}\left[\phi(X)\phi(X)^{\top}\right]$.

### 3.2 MATRIX-BASED CONDITIONAL ENTROPY AND MUTUAL INFORMATION

In the previous subsection, we reviewed the standard definition of order-$\alpha$ matrix-based entropy for PSD matrices. Here, we discuss an extension proposed by Giraldo et al. (2014) to define matrix-based conditional entropy and mutual information for two variables $X \in \mathcal{X}$ and $T \in \mathcal{T}$. For variables $X$ and $T$, we consider normalized kernel functions $k_X : \mathcal{X} \times \mathcal{X} \to \mathbb{R}$ and $k_T : \mathcal{T} \times \mathcal{T} \to \mathbb{R}$, where the kernel functions satisfy $k_X(x, x) = 1$ and $k_T(t, t) = 1$ for every input $x$ and $t$. Given the two kernel functions, Giraldo et al. (2014) define the order-$\alpha$ matrix-based joint entropy $H_\alpha(X, T)$ as the order-$\alpha$ entropy of the PSD matrix $\frac{1}{n}K_X \odot K_T$, in which $K_X$ and $K_T$ are the kernel matrices of $X$ and $T$ samples and $\odot$ denotes the entry-wise Hadamard product.

Note that the Hadamard product $K_X \odot K_T$ represents the kernel matrix of concatenated samples $[x_i, t_i]$, where we consider the kernel function $k_{X,T}([x, t], [x', t']) = k_X(x, x')k_T(t, t')$ to be the product of marginal kernel functions. This definition is sensible, since the joint similarity value, taking value over $[0, 1]$ in Gaussian kernels, is considered to be the multiplication of the similarity scores for the text and output data vectors. Then, Giraldo et al. (2014) propose defining conditional entropy $H_\alpha(X|T)$ as the difference between the joint and marginal entropy values:

$$H_\alpha(X|T) := H_\alpha(X, T) - H_\alpha(T) = H_\alpha\left(\frac{1}{n}K_X \odot K_T\right) - H_\alpha\left(\frac{1}{n}K_T\right) \tag{5}$$

Specifically, it is shown that the defined conditional entropy $H_\alpha(X|T)$ is non-negative for every normalized kernel function $k_X$ and $k_T$. Furthermore, Giraldo et al. (2014) define the matrix-based mutual information $I_\alpha(X; T)$ as the difference between the defined conditional and marginal entropy which is shown to be non-negative given normalized kernel functions $k_X$ and $k_T$:

$$I_\alpha(X; T) := H_\alpha(X) - H_\alpha(X|T) = H_\alpha\left(\frac{1}{n}K_X\right) + H_\alpha\left(\frac{1}{n}K_T\right) - H_\alpha\left(\frac{1}{n}K_X \odot K_T\right) \tag{6}$$

## 4 AN INFORMATION-THEORETIC DIVERSITY QUANTIFICATION FOR PROMPT-BASED GENERATIVE MODELS

We aim to extend the entropy-based diversity scores for unconditional generative models to conditional text-based generative models. Note that if we only evaluate the entropy score of the generated samples $x_1, \ldots, x_n$, the evaluated score does not separate the diversity contributed by different prompts from the internal diversity of the model creating varying samples for similar prompts.

To separate out the effect of diverse input prompts on the variety of the generated samples $\mathbf{x}_1, \ldots, \mathbf{x}_n$, we propose applying the conditional entropy as formulated by Giraldo et al. (2014) for a general quantum information-theoretic setting and propose the following *order-$\alpha$ Conditional-Vendi Score*:

$$\text{Conditional-Vendi}_\alpha\left(x_1, \ldots, x_n \mid t_1, \ldots, t_n\right) := \exp\left(H_\alpha\left(\frac{1}{n}K_X \odot K_T\right) - H_\alpha\left(\frac{1}{n}K_T\right)\right)$$

In the above, $K_X$ and $K_T$ denote the kernel matrices for the generated samples and input prompts, respectively. In addition to the proposed Conditional-Vendi score for measuring the internal diversity of the model, we propose the *Information-Vendi* score following the identity $I(X; T) = H(X) + H(T) - H(X, T)$ in standard information theory:

$$\text{Information-Vendi}_\alpha\left(x_1, \ldots, x_n; t_1, \ldots, t_n\right) := \exp\left(H_\alpha\left(\frac{1}{n}K_X\right) + H_\alpha\left(\frac{1}{n}K_T\right) - H_\alpha\left(\frac{1}{n}K_X \odot K_T\right)\right)$$

Note that the defined Conditional-Vendi$_\alpha$ and Information-Vendi$_\alpha$ lead to a decomposition of Vendi$_\alpha$ score (Pasarkar & Dieng, 2024) into the product of the following two terms:

$$\text{Vendi}_\alpha(x_1,\ldots,x_n) = \text{Conditional-Vendi}_\alpha(x_1,\ldots,x_n|t_1,\ldots,t_n)$$
$$\times \text{Information-Vendi}_\alpha(x_1,\ldots,x_n;t_1,\ldots,t_n).$$

In the above, Vendi$_\alpha$ measures the diversity of generated data $x_1,\ldots,x_n$, while the Conditional-Vendi$_\alpha$ score is a quantification of the internal diversity of the model that is not influenced by the variety of input prompts. Also, Information-Vendi$_\alpha$ can be viewed as a relevance score quantifying how the diversity of generated samples is statistically correlated with the diversity of text prompts. Therefore, the proposed decomposition leads to a mechanism for the internal diversity evaluation of conditional generative models.

## 5 STATISTICAL INTERPRETATION OF THE ENTROPY-BASED SCORES

In this section, we aim to statistically interpret the defined conditional diversity scores as the expectation of prompt-specific entropy $H(X|T = t)$. First, we derive the statistic estimated from empirical samples by the entropy-based scores, and then we connect the conditional entropy measure to the expectation of unconditional entropy values. According to the Schur product theorem, the Hadamard product $K_X \odot K_T$ of PSD kernel matrices $K_X$, $K_T$ will also be a PSD kernel matrix. We note that the kernel matrix corresponds to the following feature map $\phi_{X,T} : \mathcal{X} \times \mathcal{T} \to \mathbb{R}^{d_x d_t}$ where $\otimes$ denotes the Kronecker product:

$$\phi_{X,T}([x,t]) = \phi_X(x) \otimes \phi_T(t)$$

The above holds due to the identity $\langle \phi_{X,T}([x,t]), \phi_{X,T}([x',t']) \rangle = k_X(x,x')k_T(t,t')$. The following proposition formulates kernel-based conditional entropy and mutual information using $\phi_{X,T}$.

**Proposition 1.** *Consider the kernel matrices $K_X$ for samples $x_1,\ldots,x_n$ and $K_T$ for samples $t_1,\ldots,t_n$. Then, $\frac{1}{n}K_X \odot K_T$ (used for defining joint entropy $H_\alpha(X,T)$) share the same non-zero eigenvalues with the following kernel covariance matrix:*

$$\widehat{C}_{X,T} := \frac{1}{n}\sum_{i=1}^{n}\phi_{X,T}([x_i,t_i])\phi_{X,T}([x_i,t_i])^\top = \frac{1}{n}\sum_{i=1}^{n}\left[\phi_X(x_i)\phi_X(x_i)^\top\right] \otimes \left[\phi_T(t_i)\phi_T(t_i)^\top\right]$$

**Corollary 1.** *Consider the composite feature map $\phi_{X,T}$ and joint kernel covariance matrix $\widehat{C}_{X,T}$ defined above. Then, given marginal kernel covariance matrices $\widehat{C}_X = \frac{1}{n}\sum_{i=1}^{n}\phi_X(x_i)\phi_X(x_i)^\top$, $\widehat{C}_T = \frac{1}{n}\sum_{i=1}^{n}\phi_T(t_i)\phi_T(t_i)^\top$, the following holds for the defined conditional entropy and mutual information:*

$$H_\alpha(X|T) = H_\alpha(\widehat{C}_{X,T}) - H_\alpha(\widehat{C}_T), \quad I_\alpha(X;T) = H_\alpha(\widehat{C}_X) + H_\alpha(\widehat{C}_T) - H_\alpha(\widehat{C}_{X,T})$$

*Proof.* The proof is deferred to the Appendix. Note that Conditional-Vendi$_\alpha(x_1,.,x_n|t_1,.,t_n) = \exp(H_\alpha(X|T))$ and Information-Vendi$_\alpha(x_1,.,x_n;t_1,.,t_n) = \exp(I_\alpha(X;T))$ □

Corollary 1 shows that given the underlying covariance matrices $C_X = \mathbb{E}_{x \sim P_X}[\phi_X(x)\phi_X(x)^\top]$, $C_T = \mathbb{E}_{t \sim P_T}[\phi_T(t)\phi_T(t)^\top]$, and $C_{X,T} = \mathbb{E}_{(x,t) \sim P_{X,T}}[\phi_{X,T}([x,t])\phi_{X,T}([x,t])^\top]$, the defined entropy-based scores converge to the following statistics when the sample size $n$ tends to infinity:

$$\widetilde{H}_\alpha(X|T) = H_\alpha(C_{X,T}) - H_\alpha(C_T), \quad \widetilde{I}_\alpha(X;T) = H_\alpha(C_X) + H_\alpha(C_T) - H_\alpha(C_{X,T}).$$

Note that the entropy-based statistic $H_\alpha(C_X)$ represents the statistic estimated by the logarithm of the Vendi score defined in (Friedman & Dieng, 2023).

Next, we prove that for a mixture text distribution $P_T$ where the text variable follows random mode $G \in \{1,\ldots,m\}$, the defined conditional entropy score aggregates the expectation of the unconditional entropy score $H(X|G = i)$ over the $m$ text modes $1,\ldots,m$.

**Theorem 1.** *Consider the Gaussian kernel with bandwidth $\sigma$. Suppose $T$ follows a mixture distribution $\sum_{i=1}^{m} \omega_i P_{T,i}$ where $\omega_i$ denotes the weight of the $i$th component $P_{T,i}$ with mean vector $\mu_i$ and total variance $\mathbb{E}_{T \sim P_{T,i}}[\|T - \mu_i\|_2^2] = \sigma_i^2$. Given the aggregation map $f(z) = \exp((1 - \alpha)z)$, for every order $\alpha \geq 2$, the matrix-based order-$\alpha$ conditional entropy satisfies the following inequality where $g(z) = \frac{\alpha}{\alpha-1} \log\left(\frac{1}{1-z/\|\boldsymbol{\omega}\|_\alpha}\right)$ is an increasing scalar function with $g(0) = 0$:*

$$\left| \widetilde{H}_\alpha(X|T) - f^{-1}\Big( \mathbb{E}_{I \sim \boldsymbol{\omega}^\alpha}\Big[ f\big( \widetilde{H}_\alpha(X|G=I)\big)\Big]\Big)\right| \leq 2\, g\Big( 32 \sum_{i=1}^{k} \omega_i \Big[ \frac{\sigma_i^2}{\sigma^2} + \sum_{j=1}^{i-1} \exp\Big( \frac{-\|\mu_i - \mu_j\|_2^2}{\sigma^2}\Big)\Big]\Big)$$

The above theorem shows that if the text samples come from $m$ distinct modes satisfying $\frac{\|\mu_i-\mu_j\|_2}{\sigma} \gg 1$ for every $i \neq j$ and $\frac{\mathbb{E}_{P_{T,i}}[\|T-\mu_i\|_2^2]}{\sigma^2} \ll 1$, then the defined conditional entropy score $H(X|T)$ aggregates the unconditional entropy score $H(X|G=i)$ given the prompt group. Therefore, this result extends the expectation-based interpretation of Shannon conditional entropy to the matrix-based conditional entropy defined in Giraldo et al. (2014).

Based on Theorem 1, we propose a text-type-based diversity evaluation, where we restrict the evaluation of the prompt-based generative model to the prompts in the same group, i.e. the same mode in the mixture text distribution. To do this, we find the eigendirections corresponding to the text clusters by performing an eigendecomposition of text kernel matrix $\frac{1}{n} K_T = \sum_{i=1}^{n} \lambda_i v_i v_i^\top$ where $\lambda_1 \geq \cdots \lambda_n$ are the sorted eigenvalues and $v_1, \ldots, v_n$ are the sorted eigenvectors. Then, we note that the Hadamard product $\frac{1}{n} K_X \odot K_T$ used for joint entropy can be decomposed as: $\frac{1}{n} K_X \odot K_T = \sum_{i=1}^{n} \lambda_i \big( K_X \odot v_i v_i^\top\big)$. As a result, to evaluate the diversity of the generation model, we apply eigendocomposition to each $K_X \odot v_i v_i^\top$, where $v_i$ marks the samples in group $i$, and find the sample indices with the maximum entries on the principal eigenvectors of $K_X \odot v_i v_i^\top$.

# 6 NUMERICAL RESULTS

We empirically evaluated the Conditional-Vendi and Information-Vendi scores for three types of conditional generative models: 1) text-to-image, 2) text-to-video generation, and 3) image-captioning models. For text-to-image models, we tested Flux (Lab, 2024), Stable Diffusion 2.1 (Rombach et al., 2022), Stable Diffusion XL (Podell et al., 2024), GigaGAN (Kang et al., 2023), Kandinsky (Razzhigaev et al., 2023), and PixArt (Chen et al., 2023b; 2024b). For prompt-based video generative models, we considered VideoCrafter1 (Chen et al., 2023a), Show-1 (Zhang et al., 2023), and Open-Sora (Zheng et al., 2024). For image-captioning models, we experimented with BLIP (Li et al., 2022), GIT (Wang et al., 2022), and GPT4o-mini (OpenAI, 2024).

**Embeddings used in the evaluation of generative models.** Unlike standard embedding-based scores for text-to-image models such as CLIPScore (Hessel et al., 2021), which require the same embedding model for the text and generated image, the definitions of Conditional-Vendi and Information-Vendi allow different feature extractors for text and generated sample. In our experiments, we followed (Stein et al., 2023; Kynkäänniemi et al., 2023), to use the DINOv2 (Oquab et al., 2023) embedding for image data. For text data, we used Gemini (Team, 2024) and CLIP (Radford et al., 2021), and for video samples, following the video evaluation literature (Kim et al., 2024; Saito et al., 2020; Unterthiner et al., 2019), we used I3D (Carreira & Zisserman, 2017). To select the bandwidth parameter $\sigma$, similar to (Jalali et al., 2023), we chose the Gaussian kernel bandwidth for each type of data as the smallest $\sigma$ that ensures a variance below 0.01 in the evaluated score over independent evaluations. We observed that for image data, $\sigma \in [20, 30]$; for text data, $\sigma \in [0.1, 0.8]$; and for video data, $\sigma \in [10, 20]$ can satisfy this requirement.

**Quantifying model-induced diversity via Conditional-Vendi.** To illustrate how Conditional-Vendi correlates with the model-induced diversity, we considered a toy experiment with 10 different dog breeds from the ImageNet dataset (Deng et al., 2009) as simulated outputs for a text-to-image model. We generated two sets of prompts using GPT4o (OpenAI, 2024). In the first set, the breed of dog in the picture was not specified, while in the other one, the breed was explicitly mentioned. As shown in Figure 2, increasing the number of breeds sampled from the dataset led to the growth of the Vendi score, regardless of the text prompt. However, Conditional-Vendi only increased when the breed was not specified in the prompts, and in the second case where the breed had been included in the prompt, the score remained relatively constant, implying that the diversity in pictures

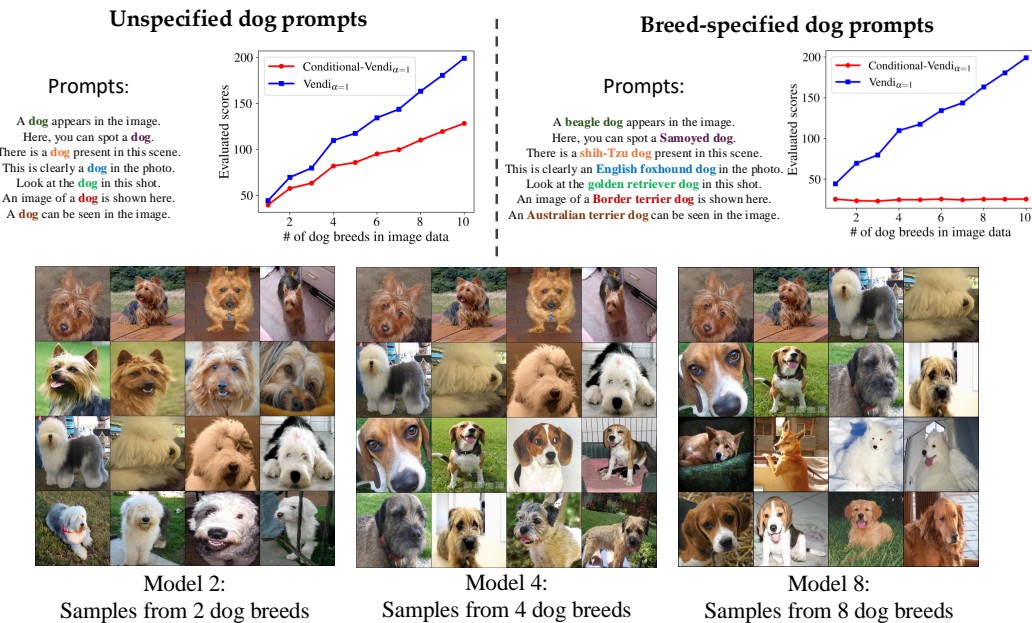

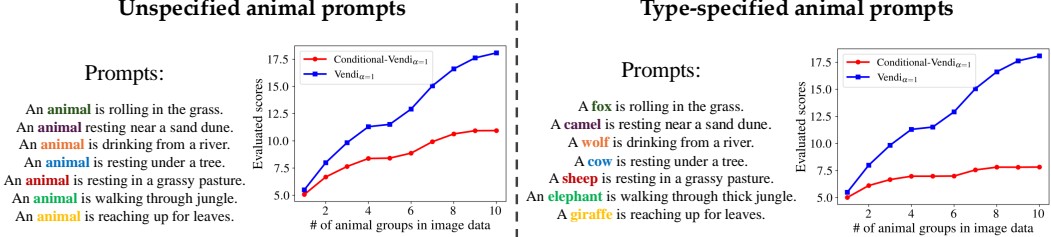

Model 2:
Samples from 2 dog breeds

Model 4:
Samples from 4 dog breeds

Model 8:
Samples from 8 dog breeds

Figure 2: Evaluated Conditional-Vendi and Vendi scores on dog samples in the ImageNet dataset. (Left Plot): we simulated prompts on "dog" pictures without specifying the dog breed, (Right Plot) we added the breed information for dogs to the prompts.

Figure 3: Evaluated Conditional-Vendi and Vendi scores on animal samples generated by Stable Diffusion-XL. (Left Plot): we do not specify the animal types in the prompt, (Right Plot) we specify the animal types in the prompt.

mostly follows the text prompt. To repeat this observation for text-to-image models, we considered 10 types of animals generated by Stable Diffusion XL, as shown in Figure 3. Similar to the previous experiment, we found that Conditional-Vendi increased at a more rapid rate when the prompts did not specify the type of animal in the picture. In contrast, when the animal types were specified in the prompts, there was only a slight increase in the Conditional-Vendi value.

**Text-to-Image Model Evaluation**: In Figure 4, we compared Flux, Stable Diffusion XL, Giga-GAN, and Kandinsky. As shown in the plot, we generated 30,000 samples using each model, and then clustered the prompts into $k$ groups, using k-means clustering on the Gemini embedding of text data, for different values of $k$. To simulate varying diversity across clusters, we assigned the image generated for the center of each cluster to all prompts within that cluster and measured the scores for different clusters. For example, when $k=2000$, we had 2,000 images for the $k$ clusters, with the text in each cluster paired with one of the images. As $k$ grows, we observed an increase in Information-Vendi, validating the fact that the images become more relevant to their prompts. Also, the Conditional-Vendi increased, as we expect the diversity of images to grow with more text clusters. Notably, while GigaGAN achieved a higher Vendi score, its Information-Vendi score was lower than that of SD-XL. This observation suggests that GigaGAN performs well at generating diverse outputs given prompts, but in terms of relevance, Flux and SD-XL are better. Our results align with conclusions made by Astolfi et al. (2024).

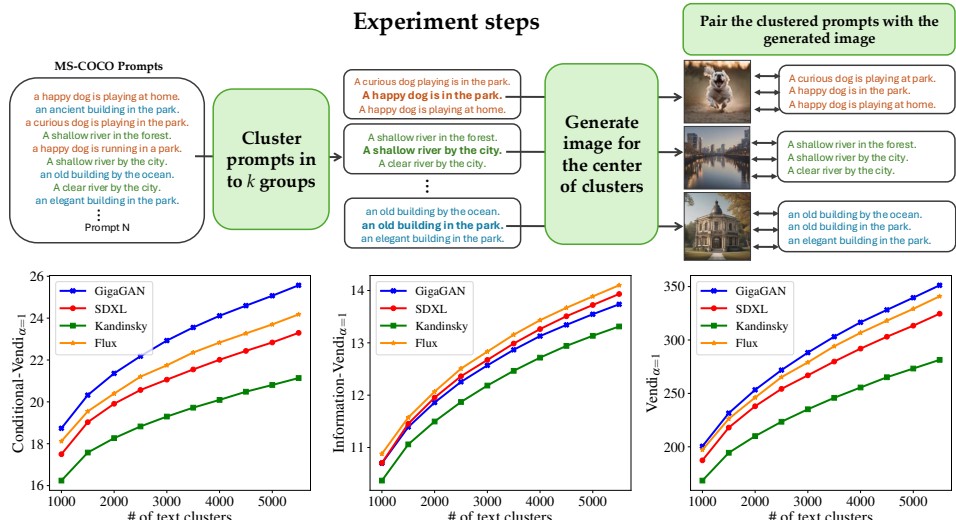

Figure 4: Comparing Conditional-Vendi and Information-Vendi of different text-to-image models.

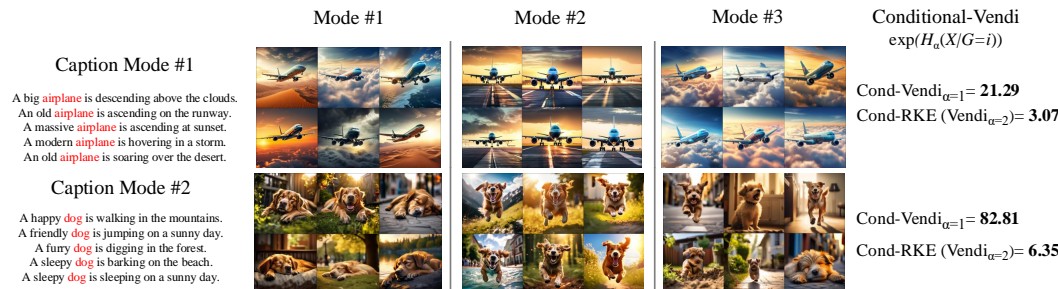

Figure 5: Quantifying image diversity for different clusters of text prompts. Images are generated using the PixArt-$\alpha$ model.

**Measuring Conditional-Vendi across prompt types.** To measure Conditional-Vendi conditioned on the prompt type, we created 5000 prompts with different categories using GPT4o and generated the corresponding images with the text-to-image models. In Figure 5, top 2 groups of PixArt-$\alpha$ in terms of conditional entropy values are shown. We observed that "dog" text-based top 3 clusters of images looked more diverse than the image clusters for "airplane"-type prompts. Also, our evaluated Conditional-Vendi score of "dog" texts was significantly higher than that of the "airplane" class. We have tested other generative models in the appendix.

**Text-to-Video Model Evaluation.** For the experiments on video data, to ensure the fairness of our evaluation, we used VBench samples (Huang et al., 2024), which generated samples belong to the 8 content categories. In Figure 6, we used VideoCrafter-1, Show-1, and Open-Sora-1.2. We observed that VideoCrafter videos look less diverse and, in some cases, may not correlate significantly with the captions, when compared to Open-Sora. Confirming this observation, the Conditional-Vendi and Information-Vendi scores were lower for VideoCrafter than those for Open-Sora.

**Image-Captioning Evaluation.** For image captioning, we used 10 classes from the ImageNet dataset as input for BLIP-2, GIT and GPT4o-mini. In Figure 7, we compared captions for the top three groups of images: gas pump, church, and cassette player. GIT generated more diverse captions compared to BLIP, which was confirmed by the Conditional-Vendi scores. On the other hand, GPT4o-mini generated longer and more detailed captions compared to GIT, which was also reflected in the evaluated Conditional-Vendi and Information-Vendi scores.

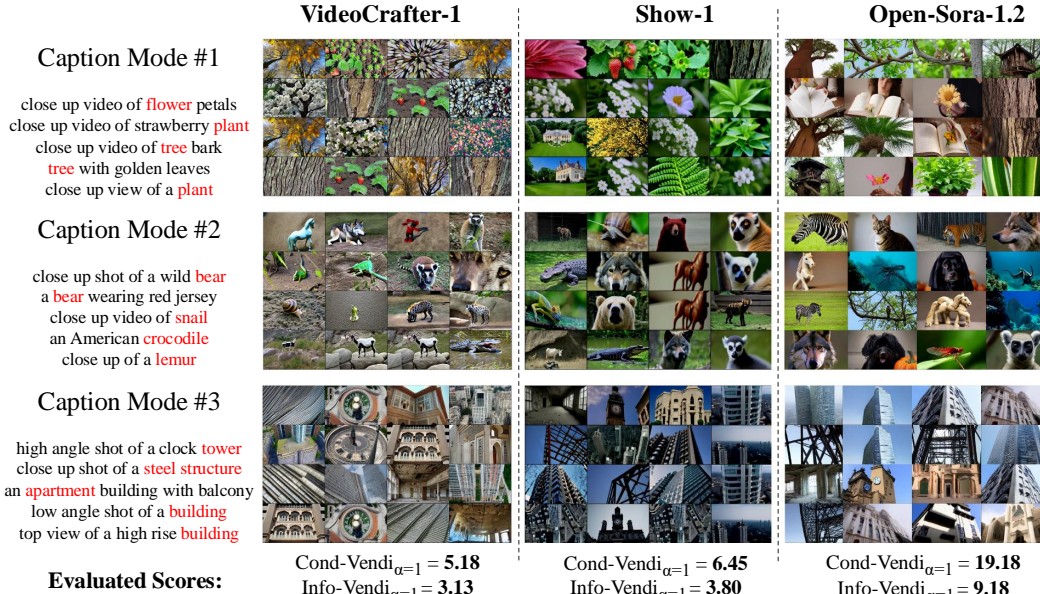

Figure 6: Measuring Conditional-Vendi and Information-Vendi for text-to-video models

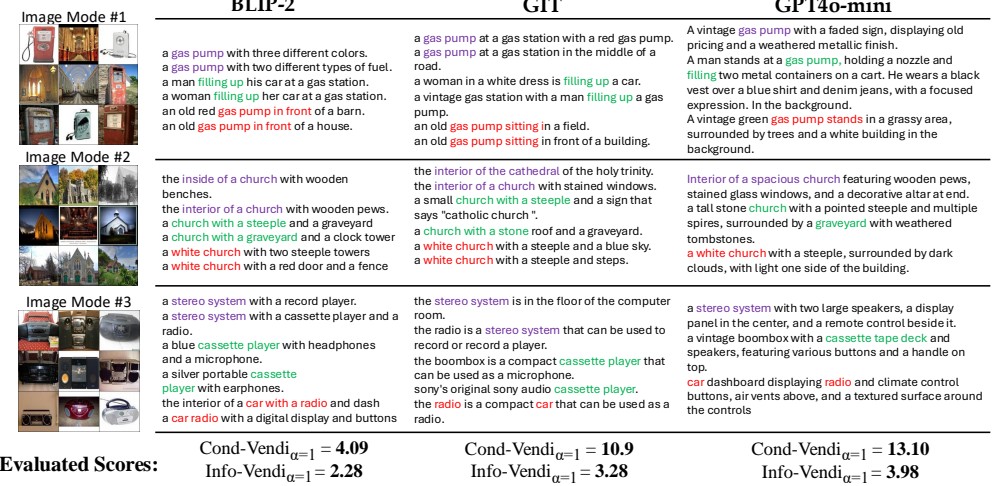

Figure 7: Conditional-Vendi and Information-Vendi of image-captioning models for 3 image types

## 7 CONCLUSION

In this work, we proposed an evaluation score to measure the internal diversity of prompt-based generative models, isolating diversity that is not induced by variations in text prompts. The proposed method is based on a decomposition of unconditional matrix-based entropy scores, Vendi and RKE, into Conditional-Vendi and Information-Vendi components. From a theoretical perspective, we derived the kernel-based statistics estimated by these scores and demonstrated their connection to the expectation of unconditional entropy values given a fixed text prompt. In our experiments, we evaluated the proposed scores in multiple settings where the ground-truth ranking of model diversity and relevance was known, showing that the scores correlate well with the ground-truth rankings. A future direction is to apply the proposed scores to quantify biases in existing models regarding sample generation across different human ethnicities and genders. Additionally, using these scores as a regularization penalty to train more diverse prompt-based models is another interesting area for further exploration.

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

# A PROOFS

## A.1 PROOF OF PROPOSITION 1

First, we observe that for every $x, x' \in \mathcal{X}$ and $t, t' \in \mathcal{T}$, the following holds:

$$
\begin{aligned}
\phi_{X,T}([x,t])^\top \phi_{X,T}([x',t']) &= \big(\phi_X(x) \otimes \phi_T(t)\big)^\top \big(\phi_X(x') \otimes \phi_T(t')\big) \\
&= \big(\phi_X(x)^\top \phi_X(x')\big) \otimes \big(\phi_T(t)^\top \phi_T(t')\big) \\
&= k_X(x,x') \otimes k_T(t,t') \\
&= k_X(x,x') k_T(t,t')
\end{aligned}
$$

Therefore, the Hadamard product of kernel matrices $\frac{1}{n} K_X \odot K_T$ can be written as

$$
\frac{1}{n} K_X \odot K_T = \frac{1}{n} \Phi_{X,T} \Phi_{X,T}^\top
$$

in terms of the matrix of samples' feature maps $\Phi_{X,T} \in \mathbb{R}^{n \times d_x d_t}$ with its $i$th row being $\phi_{X,T}([x_i, t_i])$. We observe that the matrices $\frac{1}{n} \Phi_{X,T} \Phi_{X,T}^\top$ and $\frac{1}{n} \Phi_{X,T}^\top \Phi_{X,T}$ share the same non-zero eigenvalues, that are the square of the singular values of $\Phi_{X,T}$. Therefore, $\frac{1}{n} K_X \odot K_T$ has the same non-zero eigenvalues as the following matrix

$$
\frac{1}{n} \Phi_{X,T}^\top \Phi_{X,T} = \frac{1}{n} \sum_{i=1}^n \phi_{X,T}([x_i, t_i]) \phi_{X,T}([x_i, t_i])^\top
$$

which is the defined matrix $\widehat{C}_{X,T}$. Therefore, the proof of the proposition is complete.

## A.2 PROOF OF COROLLARY 1

As we showed in Proposition 1, the Hadarmard product $\frac{1}{n} K_X \odot K_T$ shares the same non-zero eigenvalues with $\widehat{C}_{X,T}$. Also, as noted by Jalali et al. (2023), $\frac{1}{n} K_X$ and $\frac{1}{n} K_T$ have the same non-zero eigenvalues as of $\widehat{C}_X$ and $\widehat{C}_T$, respectively. Since the order-$\alpha$ matrix-based entropy is only a function of the input matrix's non-zero eigenvalues (zero eigenvalues have no impact on the entropy value), we can conclude that

$$
\begin{aligned}
H_\alpha(X|T) &:= H_\alpha\big(\frac{1}{n} K_X \odot K_T\big) - H_\alpha\big(\frac{1}{n} K_T\big) \\
&= H_\alpha(\widehat{C}_{X,T}) - H_\alpha(\widehat{C}_T),
\end{aligned}
$$

and also

$$
\begin{aligned}
I_\alpha(X;T) &:= H_\alpha\big(\frac{1}{n} K_X\big) + H_\alpha\big(\frac{1}{n} K_T\big) - H_\alpha\big(\frac{1}{n} K_X \odot K_T\big) \\
&= H_\alpha(\widehat{C}_X) + H_\alpha(\widehat{C}_T) - H_\alpha(\widehat{C}_{X,T}).
\end{aligned}
$$

## A.3 PROOF OF THEOREM 1

To prove Theorem 1, we begin by showing the following lemma.

**Lemma 1.** *Suppose that the kernel function $k$ and variable $T$ satisfy the assumptions in Theorem 1. Then, the following Frobenius norm bound holds for $C_i = \mathbb{E}[\phi_X(x)\phi_X(x)^\top | G = i]$ where $G \in \{1, \dots, m\}$ is the cluster random variable for text $T$:*

$$\left\| C_{X \otimes T} - \sum_{i=1}^m \omega_i C_i \otimes \phi_T(\mu_i)\phi_T(\mu_i)^\top \right\|_F^2 \le \frac{\sum_{i=1}^m 2\omega_i \sigma_i^2}{\sigma^2}.$$

*Proof.* To show this lemma, we define $T_i$ as a variable distributed as $P_{T|G=i}$. Then,

$$\left\| C_{X \otimes T} - \sum_{i=1}^m \omega_i C_i \otimes \phi(\mu_i)\phi(\mu_i)^\top \right\|_F^2$$

$$= \left\| \mathbb{E}[\phi_X(x)\phi_X(x)^\top \otimes \phi_T(t)\phi_T(t)^\top] - \sum_{i=1}^m \omega_i C_i \otimes \phi(\mu_i)\phi(\mu_i)^\top \right\|_F^2$$

$$= \left\| \sum_{i=1}^m \omega_i \mathbb{E}[\phi_X(x)\phi_X(x)^\top \otimes \phi_T(t)\phi_T(t)^\top | G = i] - \sum_{i=1}^m \omega_i C_i \otimes \phi(\mu_i)\phi(\mu_i)^\top \right\|_F^2$$

$$= \left\| \sum_{i=1}^m \omega_i \mathbb{E}[\phi_X(x)\phi_X(x)^\top \otimes \phi_T(t)\phi_T(t)^\top | G = i] - \sum_{i=1}^m \omega_i \mathbb{E}[\phi_X(x)\phi_X(x)^\top \otimes \phi_T(\mu_i)\phi_T(\mu_i)^\top | G = i] \right\|_F^2$$

$$= \left\| \sum_{i=1}^m \omega_i \mathbb{E}[\phi_X(x)\phi_X(x)^\top \otimes \left(\phi_T(t)\phi_T(t)^\top - \phi_T(\mu_i)\phi_T(\mu_i)^\top\right) | G = i] \right\|_F^2$$

$$\overset{(a)}{\le} \sum_{i=1}^m \omega_i \mathbb{E}\left[ \left\| \phi_X(x)\phi_X(x)^\top \otimes \left(\phi_T(t)\phi_T(t)^\top - \phi_T(\mu_i)\phi_T(\mu_i)^\top\right) \right\|_F^2 \Big| G = i \right]$$

$$\overset{(b)}{=} \sum_{i=1}^m \omega_i \mathbb{E}\left[ \left\| \phi_X(x)\phi_X(x)^\top \right\|_F^2 \left\| \phi_T(t)\phi_T(t)^\top - \phi_T(\mu_i)\phi_T(\mu_i)^\top \right\|_F^2 \Big| G = i \right]$$

$$\overset{(c)}{=} \sum_{i=1}^m \omega_i \mathbb{E}\left[ \left\| \phi_T(t)\phi_T(t)^\top - \phi_T(\mu_i)\phi_T(\mu_i)^\top \right\|_F^2 \Big| G = i \right]$$

$$\overset{(d)}{=} \sum_{i=1}^m \omega_i \mathbb{E}\left[ 2 - 2\exp\left(\frac{-\|t - \mu_i\|_2^2}{\sigma^2}\right) \Big| G = i \right]$$

$$\overset{(e)}{\le} \sum_{i=1}^m \omega_i \left[ 2 - 2\exp\left(\frac{-\mathbb{E}[\|t - \mu_i\|_2^2 | G = i]}{\sigma^2}\right) \right]$$

$$\overset{(f)}{\le} \sum_{i=1}^m \omega_i \left[ 2 - 2\exp\left(\frac{-\sigma_i^2}{\sigma^2}\right) \right]$$

$$\overset{(g)}{\le} \sum_{i=1}^m 2\omega_i \frac{\sigma_i^2}{\sigma^2}$$

In the above, (a) follows from Jensen's inequality for the convex Frobenius-norm-squared function. (b) holds because $\|A \otimes B\|_F^2 = \|A\|_F^2 \|B\|_F^2$ for every matrices $A$, $B$. (c) comes from the normalized Gaussian kernel satisfying $\langle \phi_T(t), \phi_T(t) \rangle = k(t, t) = 1$, resulting in $\|\phi_T(t)\phi_T(t)^\top\|_F^2 = \mathrm{Tr}\left(\phi_T(t)\phi_T(t)^\top \phi_T(t)\phi_T(t)^\top\right) = \mathrm{Tr}\left(\phi_T(t)\phi_T(t)^\top\right) = 1$. (d) follows from the Gaussian kernel definition, proving that $\phi_T(t)^\top \phi_T(\mu_i) = \exp\left(-\|t - \mu_i\|_2^2 / 2\sigma^2\right)$. (e) shows the application of Jensen's inequality to the concave $s(z) = 1 - \exp(-z)$. (f) holds because $s(z) = 1 - \exp(-z)$ is a monotonically increasing function. Finally, (g) follows from the inequality $1 - \exp(-z) \le z$ for every scalar $z$. Therefore, the proof is complete.

$\square$

Next, we apply the Gram–Schmidt process to $\phi_T(\mu_1), \ldots, \phi_T(\mu_m)$ to find orthogonal vectors $u_1, \ldots, u_m$. We let $u_1 = \phi_T(\mu_1)$. Then, for every $2 \leq i \leq m$, we define

$$u_i := \phi(\mu_i) - \sum_{j=1}^{i-1} \langle \phi(\mu_i), u_j \rangle u_j.$$

As a result, the following holds

$$\left\| \sum_{i=1}^{m} \omega_i C_i \otimes \phi(\mu_i)\phi(\mu_i)^\top - \sum_{i=1}^{m} \omega_i C_i \otimes u_i u_i^\top \right\|_F^2$$

$$= \left\| \sum_{i=1}^{m} \omega_i C_i \otimes \left( \phi(\mu_i)\phi(\mu_i)^\top - u_i u_i^\top \right) \right\|_F^2$$

$$\overset{(h)}{\leq} \sum_{i=1}^{m} \omega_i \left\| C_i \otimes \left( \phi(\mu_i)\phi(\mu_i)^\top - u_i u_i^\top \right) \right\|_F^2$$

$$= \sum_{i=1}^{m} \omega_i \left\| C_i \right\|_F^2 \left\| \phi(\mu_i)\phi(\mu_i)^\top - u_i u_i^\top \right\|_F^2$$

$$\overset{(i)}{\leq} \sum_{i=1}^{m} \omega_i \left\| \phi(\mu_i)\phi(\mu_i)^\top - u_i u_i^\top \right\|_F^2$$

$$\overset{(j)}{=} \sum_{i=1}^{m} \omega_i \left( 1 + \|u_i\|^4 - 2\left( u_i^\top \phi_T(\mu_i) \right)^2 \right)$$

$$\leq \sum_{i=1}^{m} \omega_i \left( 2 - 2\left( u_i^\top \phi(\mu_i) \right)^2 \right)$$

$$= 2 \sum_{i=1}^{m} \omega_i \left( 1 + u_i^\top \phi(\mu_i) \right) \left( 1 - u_i^\top \phi(\mu_i) \right)$$

$$\overset{(k)}{\leq} 4 \sum_{i=1}^{m} \sum_{j=1}^{i-1} \omega_i \exp\left( \frac{-\|\mu_i - \mu_j\|_2^2}{\sigma^2} \right).$$

Here, (h) follows from the application of Jensen's inequality for the convex Frobenius-norm-squared. (i) holds since the text kernel is normalized and $\langle \phi_X(x), \phi_X(x) \rangle = k_X(x, x) = 1$, and therefore $\|C_i\|_F \leq \mathbb{E}[\|\phi_X(x)\|_2^2] = 1$. (j) follows from the expansion $\|uu^\top - vv^\top\|_F^2 = \|u\|_2^4 + \|v\|_2^4 - 2\langle u, v \rangle^2$. (k) holds because $u_i^\top \phi_T(\mu_i) \leq 1$ and

$$u_i^\top \phi_T(\mu_i) = 1 - \sum_{j=1}^{i-1} \langle \phi_T(\mu_i), u_j \rangle^2 \geq 1 - \sum_{j=1}^{i-1} \exp\left( \frac{-\|\mu_i - \mu_j\|_2^2}{\sigma^2} \right).$$

Since we know that for every matrices $A, B \in \mathbb{R}^{d \times d}$, $\|A + B\|_F^2 \leq 2\|A\|_F^2 + 2\|B\|_F^2$, the above results show that

$$\left\| C_{X \otimes T} - \sum_{i=1}^{m} \omega_i C_i \otimes u_i u_i^\top \right\|_F^2 \leq \sum_{i=1}^{m} 4\omega_i \frac{\sigma_i^2}{\sigma^2} + \sum_{i=2}^{m} \sum_{j=1}^{i-1} 8\omega_i \exp\left( \frac{-\|\mu_i - \mu_j\|_2^2}{\sigma^2} \right).$$

As a result, the Hoffman-Wielandt inequality shows that for the sorted eigenvalues vector $\boldsymbol{\lambda}$ of $C_{X \otimes T}$ and sorted eigenvalues vector $\widetilde{\boldsymbol{\lambda}}$ of $\sum_{i=1}^{m} \omega_i C_i \otimes u_i u_i^\top$ the following holds:

$$\left\| \boldsymbol{\lambda} - \widetilde{\boldsymbol{\lambda}} \right\|_2^2 \leq \left\| C_{X \otimes T} - \sum_{i=1}^{m} \omega_i C_i \otimes u_i u_i^\top \right\|_F^2$$

$$\leq \sum_{i=1}^{m} 4\omega_i \frac{\sigma_i^2}{\sigma^2} + \sum_{i=2}^{m} \sum_{j=1}^{i-1} 8\omega_i \exp\left( \frac{-\|\mu_i - \mu_j\|_2^2}{\sigma^2} \right).$$

Since $u_1, \ldots, u_m$ are orthogonal vectors, the definition of Kronecker product implies that the eigenvalues of $\sum_{i=1}^{m} \omega_i C_i \otimes u_i u_i^{\top}$ will be the union of the eigenvalues of $\omega_i C_i \otimes u_i u_i^{\top}$ over $i \in \{1, \ldots, m\}$. On the other hand, we know that the non-zero eigenvalues of $\omega_i C_i \otimes u_i u_i^{\top}$ will be equal to the factor $\omega_i \|u_i\|_2^2$ times the eigenvalues of $C_i$. Also, we know that $1 \geq \|u_i\|_2^2 \geq 1 - 2 \sum_{j=1}^{i-1} \exp(-\frac{\|\mu_i - \mu_j\|_2^2}{\sigma^2})$. Consequently, we can show that for vector $\widehat{\lambda}_{x \otimes t} = \text{Union}\big(\omega_i \text{Eigs}(C_i) : i \in \{1, \ldots, m\}\big)$, we have the following for every $\alpha \geq 2$ and defined increasing function $g$ in Theorem 1

$$
\begin{aligned}
\left| \widetilde{H}_\alpha(X, T) - \frac{1}{1-\alpha} \log\big(\|\widehat{\lambda}_{x \otimes t}\|_\alpha^\alpha\big) \right| &\leq g\big(\|\widetilde{\lambda}_{x \otimes t}\|_\alpha - \|\widehat{\lambda}_{x \otimes t}\|_\alpha\big) \\
&\leq g\big(\|\text{sort}(\widetilde{\lambda}_{x \otimes t}) - \text{sort}(\widehat{\lambda}_{x \otimes t})\|_\alpha\big) \\
&\leq g\big(\|\text{sort}(\widetilde{\lambda}_{x \otimes t}) - \text{sort}(\widehat{\lambda}_{x \otimes t})\|_2\big) \\
&\leq g\Big(\sum_{i=1}^{m} 4\omega_i \frac{\sigma_i^2}{\sigma^2} + \sum_{i=2}^{m} \sum_{j=1}^{i-1} 16\omega_i \exp\Big(\frac{-\|\mu_i - \mu_j\|_2^2}{\sigma^2}\Big)\Big).
\end{aligned}
$$

Note that the above proof holds for every marginal distribution on $X$, and we choose a deterministic constant $X = \mathbf{0}$, then the joint entropy reduces to the marginal entropy and the above inequality also shows the following:

$$
\left| \widetilde{H}_\alpha(T) - \frac{1}{1-\alpha} \log\big(\|[\omega_1, \ldots, \omega_m]\|_\alpha^\alpha\big) \right| \leq g\Big(\sum_{i=1}^{m} 4\omega_i \frac{\sigma_i^2}{\sigma^2} + \sum_{i=2}^{m} \sum_{j=1}^{i-1} 16\omega_i \exp\Big(\frac{-\|\mu_i - \mu_j\|_2^2}{\sigma^2}\Big)\Big).
$$

Therefore, following the Triangle inequality and the definition $\widetilde{H}_\alpha(X|T) = \widetilde{H}_\alpha(X, T) - \widetilde{H}_\alpha(T)$, the previous two inequalities prove that

$$
\left| \widetilde{H}_\alpha(X|T) - \Big(\frac{1}{1-\alpha} \log\big(\|\widehat{\lambda}_{x \otimes t}\|_\alpha^\alpha\big) - \frac{1}{1-\alpha} \log\big(\|[\omega_1, \ldots, \omega_m]\|_\alpha^\alpha\big)\Big) \right|
$$
$$
\leq 2g\Big(\sum_{i=1}^{m} 4\omega_i \frac{\sigma_i^2}{\sigma^2} + \sum_{i=2}^{m} \sum_{j=1}^{i-1} 16\omega_i \exp\Big(\frac{-\|\mu_i - \mu_j\|_2^2}{\sigma^2}\Big)\Big).
$$

On the other hand, we can simplify the above expression as

$$
\begin{aligned}
&\frac{1}{1-\alpha} \log\big(\|\widehat{\lambda}_{x \otimes t}\|_\alpha^\alpha\big) - \frac{1}{1-\alpha} \log\big(\|[\omega_1, \ldots, \omega_m]\|_\alpha^\alpha\big) \\
&= \frac{1}{1-\alpha} \log\Big(\sum_{i=1}^{m} \omega_i^\alpha \|\lambda_{C_i}\|_\alpha^\alpha\Big) - \frac{1}{1-\alpha} \log\Big(\sum_{i=1}^{m} \omega_i^\alpha\Big) \\
&= \frac{1}{1-\alpha} \log\Big(\sum_{i=1}^{m} \frac{\omega_i^\alpha}{\sum_{j=1}^{m} \omega_j^\alpha} \|\lambda_{C_i}\|_\alpha^\alpha\Big)
\end{aligned}
$$

Note that the definition $f_\alpha(t) = \exp((1-\alpha)t)$ implies that $f_\alpha^{-1}(z) = \frac{1}{1-\alpha} \log(z)$, which connects to the entropy definition as $H(X|G = i) = f_\alpha^{-1}(\|\lambda_{C_i}\|_\alpha^\alpha)$. As a result, we can combine the previous two equations and complete the proof as:

$$
\left| \widetilde{H}_\alpha(X|T) - f_\alpha^{-1}\Big(\sum_{i=1}^{m} \frac{\omega_i^\alpha}{\sum_{j=1}^{m} \omega_j^\alpha} f_\alpha\big(\widetilde{H}_\alpha(X|G = i)\big)\Big) \right|
$$
$$
\leq 2g\Big(\sum_{i=1}^{m} 4\omega_i \frac{\sigma_i^2}{\sigma^2} + \sum_{i=2}^{m} \sum_{j=1}^{i-1} 16\omega_i \exp\Big(\frac{-\|\mu_i - \mu_j\|_2^2}{\sigma^2}\Big)\Big).
$$

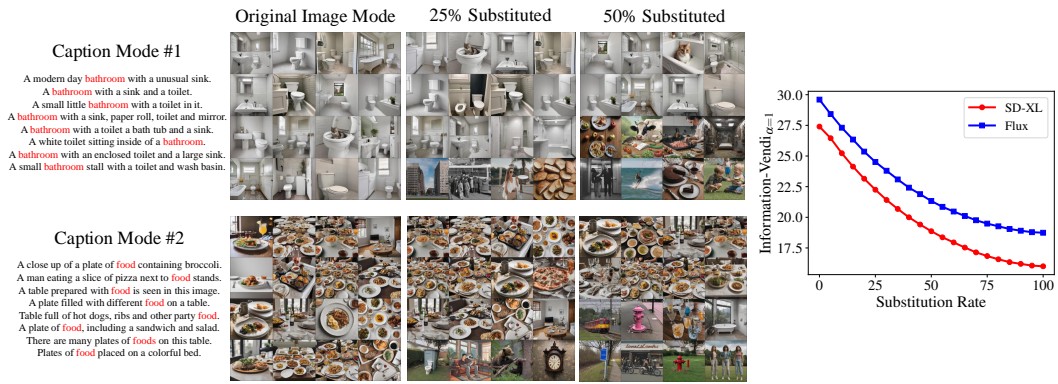

Figure 8: Substituting images generated from models trained on MS-COCO dataset.

# B  ADDITIONAL NUMERICAL RESULTS

## B.1  CORRELATION BETWEEN PROMPTS AND GENERATED OUTPUT

To measure the correlation between text and image using Information-Vendi, we used MS-COCO captions to generate images with Stable Diffusion XL and Flux. We gradually substituted the generated images with random ones for the same prompts at different substitution rates. As the substitution rate increased, the correlation between the text and image pairs decreased. In Figure 8, we measured Information-Vendi at various substitution rates and observed that as the substitution rate increased, Information-Vendi decreased, demonstrating that our score can successfully measure the correlation between text and image. Unlike other correlation metrics, such as CLIPScore, which require the same embedding for both text and image, our method places no such restriction. This allows for the use of different embeddings for text and image. Furthermore, our approach can be easily generalized to other conditional models, such as text-to-text or text-to-video generation.

## B.2  MEASURING CONDITIONAL-VENDI ACROSS PROMPT TYPES

In this section, we conducted additional experiments similar to those in Figure 5. We created 5,000 prompts across different categories using GPT4o and generated corresponding images with text-to-image models. We reported Conditional-Vendi for the top 3 groups in the text data on PixArt-$\alpha$, Stable Diffusion XL text-to-image generative models.

As shown in Figure 9, Figure 10 and Figure 11, we observed the same behavior during these experiments: the Conditional-Vendi score for "dog" prompts was significantly higher than for the "airplane" and "sofa" categories. This observation suggests that the outputs of generative models are unbalanced when presented with different groups of text prompts.

## B.3  QUANTIFYING MODEL-INDUCED DIVERSITY VIA CONDITIONAL-VENDI.

In this section, we provided a more detailed version of Figure 3. As shown in Figure 12, we found that Conditional-Vendi increased at a more rapid rate when the prompts did not specify the type of animal in the picture. In contrast, when the animal types were specified in the prompts, there was only a slight increase in the Conditional-Vendi score.

## B.4  ADDITIONAL NUMERICAL EVALUATION OF THE CONDITIONAL-VENDI SCORE

To further experiment the correlation between the intrinsic model diversity and the defined Conditional-Vendi score, we have performed experiments of quantifying the diversity scores for unspecified and type-specified prompts when generating data from standard text-to-image models. To conduct an extensive evaluation of the Conditional-Vendi score, we performed the experiments on the nine combinations of three category types, 1) animals, 2) fruits, 3) objects, and three SOTA text-to-image generation models SDXL, Kandinsky, and PixArt-$\Sigma$ Chen et al. (2024a). In each

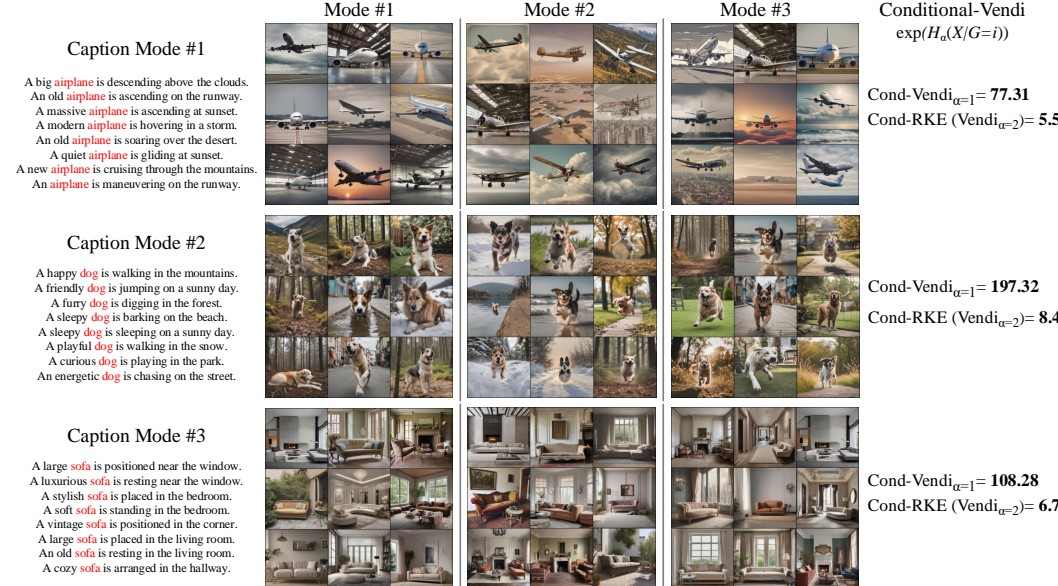

Figure 9: Quantifying image diversity for different clusters of text prompts. Images are generated using the Stable Diffusion XL model.

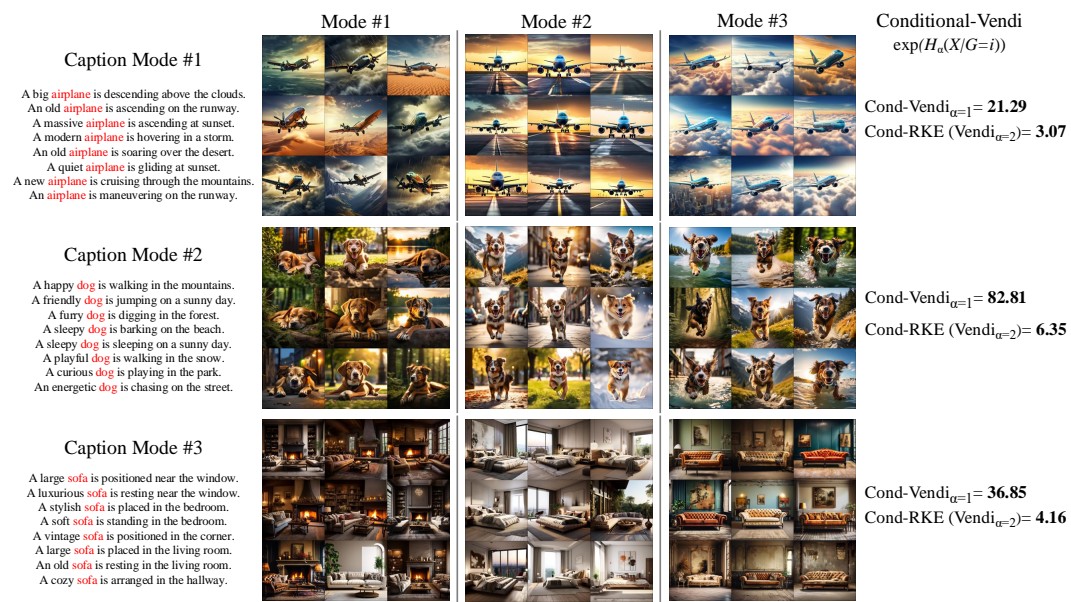

Figure 10: Quantifying image diversity for different clusters of text prompts. Images are generated using the PixArt-$\alpha$ model.

of the nine experiments, we generated prompts on 10 different types related to each category and created image samples by inputting the prompts to the text-to-image model. In each experiment, we simulated 10 prompt-based generative models by considering image samples from $j$ types for $j \in \{1, \ldots, 10\}$.

In addition, given the original prompts specifying the type of category in the image, we compared the Vendi and Conditional-Vendi scores among the three text-to-image models. Figures ??? show the comparison between the scores of the three models, which suggest the higher intrinsic diversity measure by Conditional-Vendi for the SD-XL and PixArt-$\Sigma$ models.

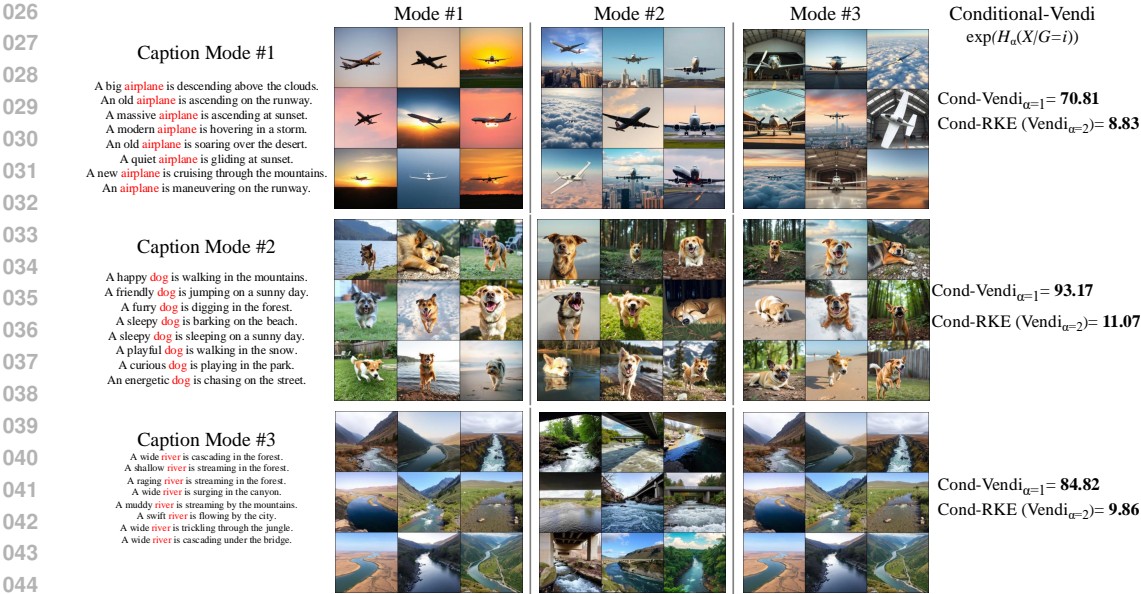

Figure 11: Quantifying image diversity for different clusters of text prompts. Images are generated using the Flux model.

### B.5 CORRELATION BETWEEN GROUNDTRUTH-CLUSTER-VENDI AND CONDITIONAL-VENDI SCORES

To validate the theoretical connection between the Vendi and Conditional-Vendi scores, we performed an experiment and evaluated a baseline metric called GroundTruth-Cluster-Vendi score. To measure the GroundTruth-Cluster-Vendi score, we utilize the side knowledge of the ground-truth clusters of the input prompts and then compute and average the regular Vendi scores for the data generated within each cluster. Mathematically, given $t$ sample cluster sets in $\mathcal{S} = \{S_1, \ldots, S_t\}$, which partition the input text indices $\{1, \ldots, n\}$, we define the Cluster-Vendi score as follows, where $|S_j|$ denotes the cardinality of subset $S_j$:

$$\text{Cluster-Vendi}\big(x_1, \ldots, x_n \,|\, \mathcal{S}\big) \;:=\; \sum_{i=1}^{t} \frac{|S_i|}{n} \cdot \text{Vendi}\Big(\big\{x_j \,:\, j \in S_i\big\}\Big).$$

Note that the above definition requires the knowledge of the clusters, which could be given by an oracle in the case of the GroundTruth-Cluster-Vendi score, or computed by a clustering algorithm such as K-Means to obtain the KMeans-Cluster-Vendi score. Observe that given the knowledge of the clusters revealed by an oracle, the GroundTruth-Cluster-Vendi score is a sensible definition of internal model diversity, which, as shown in Theorem 1, is expected to correlate with our defined Conditional-Vendi score.

In the numerical settings of the previous section, where we know the ground-truth clusters based on the type of animal, fruit, or object in the texts, we computed the value of the GroundTruth-Cluster-Vendi score and compared it with the evaluated Conditional-Vendi score. As demonstrated in Figures 22, the two diversity scores, Conditional-Vendi and GroundTruth-Cluster-Vendi, highly correlate for the ten simulated generative models in the experiments.

However, note that in a real-world scenario, we do not have access to the ground-truth clusters. To estimate the score, we should use a clustering algorithm such as K-Means to find the clusters and compute the Cluster-Vendi score. We note that the optimization problem addressed by standard clustering algorithms represents a challenging non-convex optimization, which, depending on the algorithm's initial point, could converge to different solutions. Our numerical results with the K-Means clustering algorithm in Figure 23 also demonstrated these clustering challenges and, in several cases, failed to find the ground-truth clusters with high accuracy.

# Type-specified animal prompts

### Prompts:

A **fox** is rolling in the grass.
A **camel** is resting near a sand dune.
A **wolf** is drinking from a river.
A **cow** is resting under a tree.
A **sheep** is resting in a grassy pasture.
An **elephant** is walking through thick jungle.
A **giraffe** is reaching up for leaves.

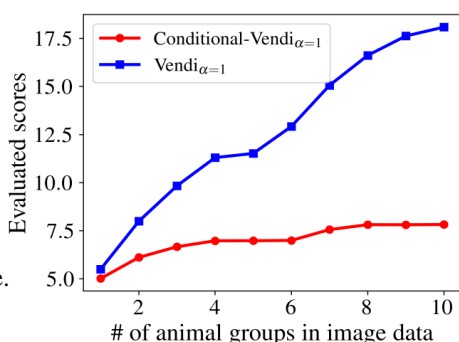

# Unspecified animal prompts

### Prompts:

An **animal** is rolling in the grass.
An **animal** resting near a sand dune.
An **animal** is drinking from a river.
An **animal** is resting under a tree.
An **animal** is resting in a grassy pasture.
An **animal** is walking through jungle.
An **animal** is reaching up for leaves.

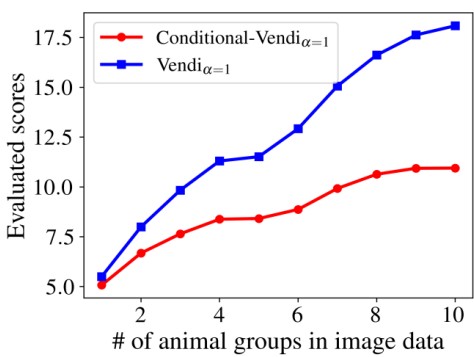

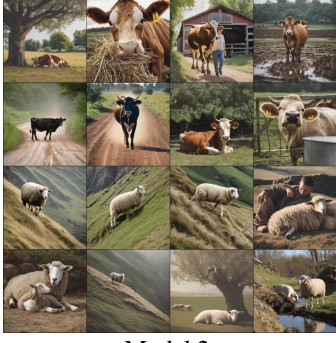 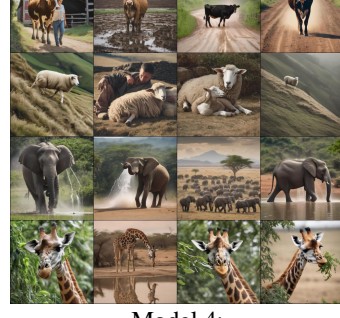 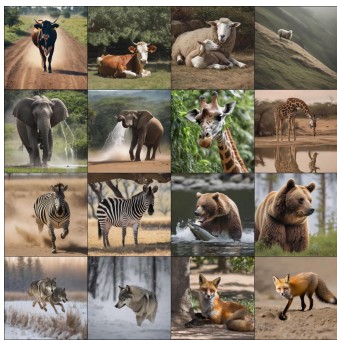

| Model 2: | Model 4: | Model 8: |
| Samples from 2 animal groups | Samples from 4 animal groups | Samples from 8 animal groups |

Figure 12: Comparing Conditional-Vendi with Vendi on different animal groups generated by Stable Diffusion-XL.

## B.6 ALGORITHM FOR COMPUTING CONDITIONAL-VENDI AND INFORMATION-VENDI

In this section, we present the algorithm to compute the Conditional-Vendi and Information-Vendi scores. Using the definition provided in Section 4, combined with the entropy definition in equation 3, we calculate the Conditional-Vendi score. The steps are outlined in Algorithm 1.

## B.7 QUALITATIVE RESULTS FOR GENERATIVE MODELS TRAINED ON MS-COCO DATASET

In this section, we provide images and prompts corresponding to Figure 4. Figure 25 illustrates three clusters obtained by applying KMeans to cluster MS-COCO validation set prompts into 1000 clusters. The images are presented for four generative models. Comparing the prompts with the generated images reveals that FLUX exhibits the highest alignment between text and image, while

# Type-specified animal prompts

Prompts:

A **fox** is rolling in the grass.
A **camel** is resting near a sand dune.
A **wolf** is drinking from a river.
A **cow** is resting under a tree.
A **sheep** is resting in a grassy pasture.
An **elephant** is walking through thick jungle
A **giraffe** is reaching up for leaves.

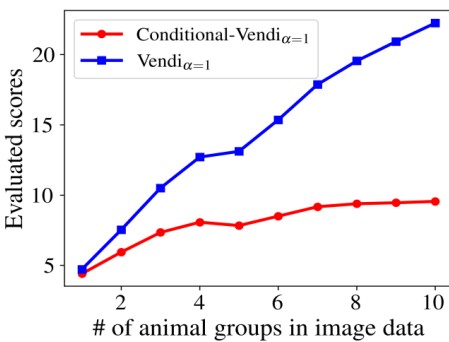

# Unspecified animal prompts

Prompts:

An **animal** is rolling in the grass.
An **animal** resting near a sand dune.
An **animal** is drinking from a river.
An **animal** is resting under a tree.
An **animal** is resting in a grassy pasture.
An **animal** is walking through jungle.
An **animal** is reaching up for leaves.

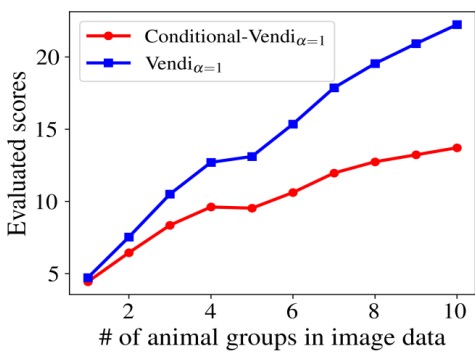

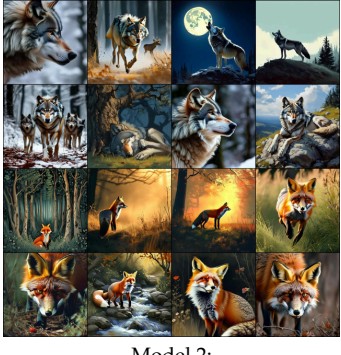

Model 2:
Samples from 2 animal groups

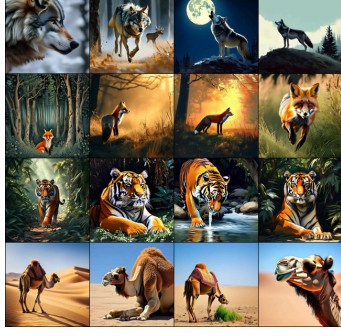

Model 4:
Samples from 4 animal groups

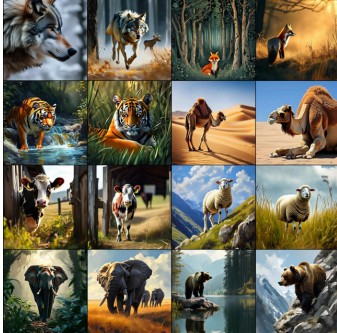

Model 8:
Samples from 8 animal groups

Figure 13: Comparing Conditional-Vendi with Vendi on different animal groups generated by PixArtΣ.

GigaGAN demonstrates greater diversity but misses some features of the prompts. These observations are further supported by the Conditional-Vendi and Information-Vendi metrics.

## B.8 EFFECT OF BANDWIDTH ON CONDITIONAL-VENDI AND INFORMATION-VENDI

To further investigate the effect of bandwidth on Conditional-Vendi and Information-Vendi, we began by selecting the image bandwidth similar to prior works Friedman & Dieng (2023); Ospanov et al. (2024). We then measured and plotted the scores using varying text kernel bandwidths. Figure 26 demonstrates consistent rankings of the four models across different bandwidth parameters. The results indicate that as the kernel bandwidth increases, the number of text clusters increases, leading to a decrease in the Information-Vendi value.

## Type-specified animal prompts

Prompts:

A **fox** is rolling in the grass.
A **camel** is resting near a sand dune.
A **wolf** is drinking from a river.
A **cow** is resting under a tree.
A **sheep** is resting in a grassy pasture.
An **elephant** is walking through thick jungle
A **giraffe** is reaching up for leaves.

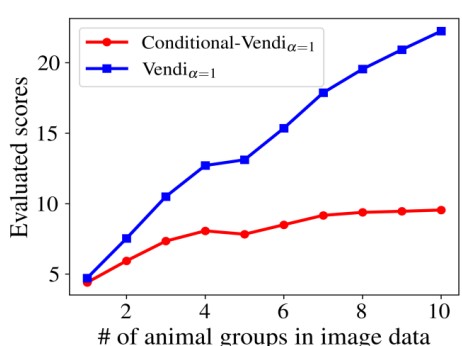

## Unspecified animal prompts

Prompts:

An **animal** is rolling in the grass.
An **animal** resting near a sand dune.
An **animal** is drinking from a river.
An **animal** is resting under a tree.
An **animal** is resting in a grassy pasture.
An **animal** is walking through jungle.
An **animal** is reaching up for leaves.

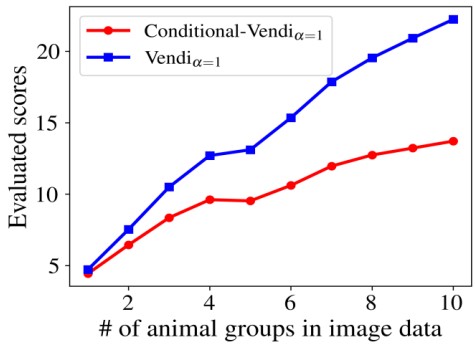

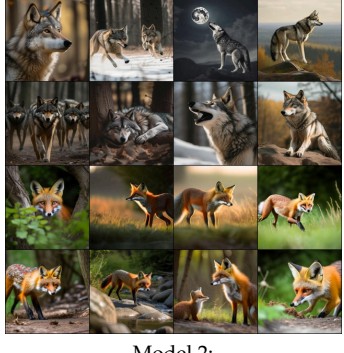
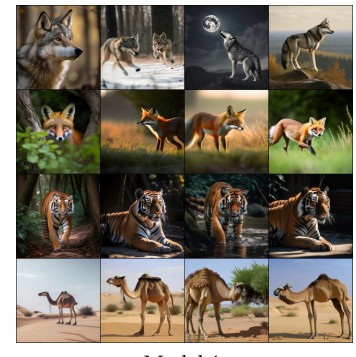
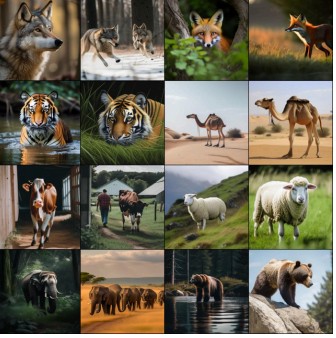

| Model 2: | Model 4: | Model 8: |
| Samples from 2 animal groups | Samples from 4 animal groups | Samples from 8 animal groups |

Figure 14: Comparing Conditional-Vendi with Vendi on different animal groups generated by Kandinsky.

# Type-specified object prompts

## Prompts:

A **Chair** is placed under a sprawling tree.
A **Sofa** is glowing in the light of a nearby fire.
A **Book** is balancing on the edge of a table.
A **Clock** is positioned in an office setup.
A **Lamp** is sitting under a hanging light bulb.
A **laptop** is half-hidden behind a stack of boxes.
A **car** is in the corner of a large warehouse.
The **cup** is precariously balanced on rocks.

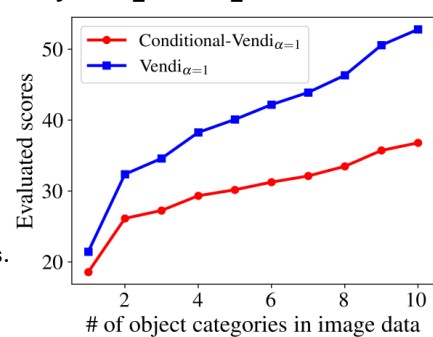

# Unspecified object prompts

## Prompts:

An **object** is placed under a sprawling tree.
An **object** is glowing in the light of a nearby fire.
An **object** is balancing on the edge of a table.
An **object** is positioned in an office setup.
An **object** is sitting under a hanging light bulb.
An **object** is half-hidden behind a stack of boxes.
An **object** is in the corner of a large warehouse.
The **object** is precariously balanced on rocks.

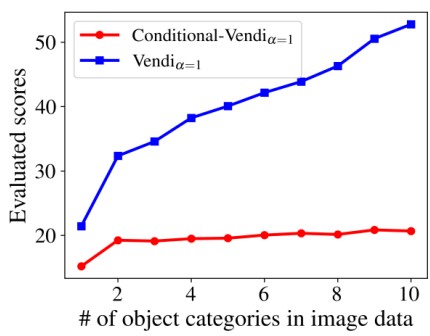

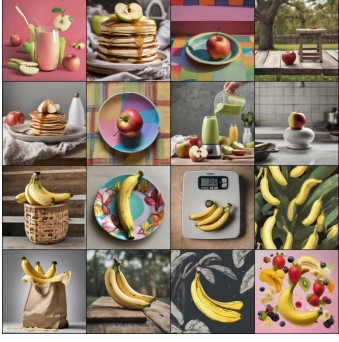
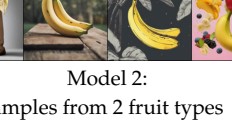
Model 2:
Samples from 2 fruit types

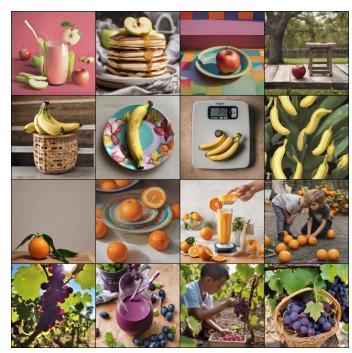
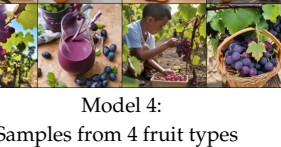
Model 4:
Samples from 4 fruit types

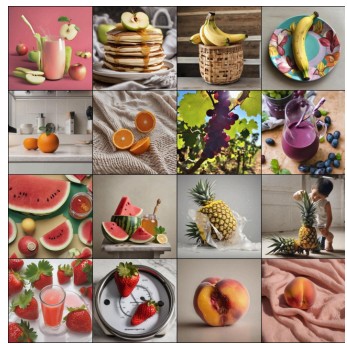
Model 8:
Samples from 8 fruit types

Figure 15: Comparing Conditional-Vendi with Vendi on different fruit types generated by Stable Diffusion-XL.

# Type-specified fruit prompts

## Prompts:

An **apple** is next to a cold glass of fresh juice.
A **banana** is being sliced with a sharp knife.
The **watermelon** is blended into a smoothie.
The **pineapple** is falling out of a grocery bag.
A **strawberry** is being washed.
The **peach** is sitting on a kitchen countertop.
A **cherry** is being sliced with a knife.
A **mango** is sitting on a colorful plate.

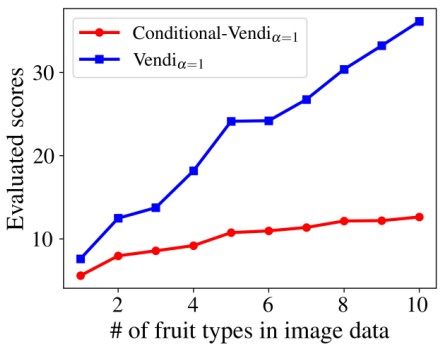

# Unspecified fruit prompts

## Prompts:

A **fruit** is next to a cold glass of fresh juice.
A **fruit** is being sliced with a sharp knife.
The **fruit** is blended into a smoothie.
The **fruit** is falling out of a grocery bag.
A **fruit** is being washed.
The **fruit** is sitting on a kitchen countertop.
A **fruit** is being sliced with a knife.
A **fruit** is sitting on a colorful plate.

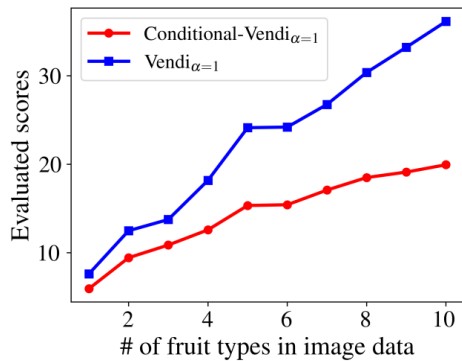

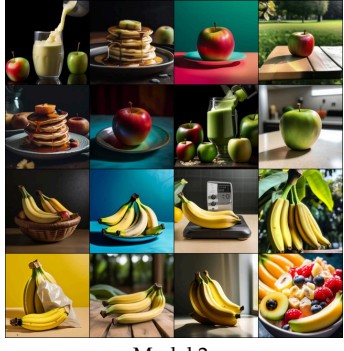

Model 2:
Samples from 2 fruit types

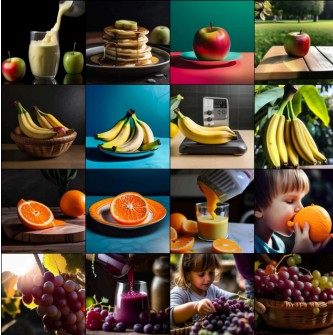

Model 4:
Samples from 4 fruit types

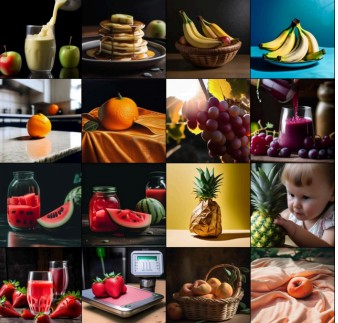

Model 8:
Samples from 8 fruit types

Figure 16: Comparing Conditional-Vendi with Vendi on different fruit types generated by Kandinsky.

# Type-specified fruit prompts

## Prompts:

An **apple** is next to a cold glass of fresh juice.
A **banana** is being sliced with a sharp knife.
The **watermelon** is blended into a smoothie.
The **pineapple** is falling out of a grocery bag.
A **strawberry** is being washed.
The **peach** is sitting on a kitchen countertop.
A **cherry** is being sliced with a knife.
A **mango** is sitting on a colorful plate.

# Unspecified fruit prompts

## Prompts:

A **fruit** is next to a cold glass of fresh juice.
A **fruit** is being sliced with a sharp knife.
The **fruit** is blended into a smoothie.
The **fruit** is falling out of a grocery bag.
A **fruit** is being washed.
The **fruit** is sitting on a kitchen countertop.
A **fruit** is being sliced with a knife.
A **fruit** is sitting on a colorful plate.

Model 2:
Samples from 2 fruit types

Model 4:
Samples from 4 fruit types

Model 8:
Samples from 8 fruit types

Figure 17: Comparing Conditional-Vendi with Vendi on different fruit types generated by PixArt-$\Sigma$.

## Type-specified object prompts

### Prompts:

A **Chair** is placed under a sprawling tree.
A **Sofa** is glowing in the light of a nearby fire.
A **Book** is balancing on the edge of a table.
A **Clock** is positioned in an office setup.
A **Lamp** is sitting under a hanging light bulb.
A **laptop** is half-hidden behind a stack of boxes.
A **car** is in the corner of a large warehouse.
The **cup** is precariously balanced on rocks.

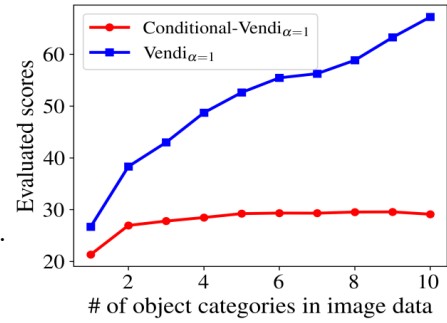

## Unspecified object prompts

### Prompts:

An **object** is placed under a sprawling tree.
An **object** is glowing in the light of a nearby fire.
An **object** is balancing on the edge of a table.
An **object** is positioned in an office setup.
An **object** is sitting under a hanging light bulb.
An **object** is half-hidden behind a stack of boxes.
An **object** is in the corner of a large warehouse.
The **object** is precariously balanced on rocks.

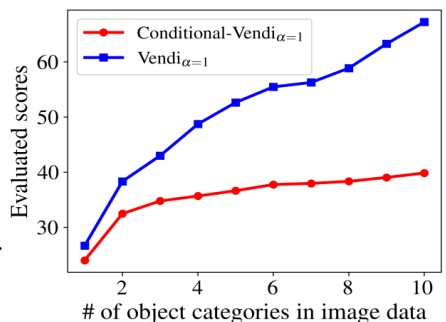

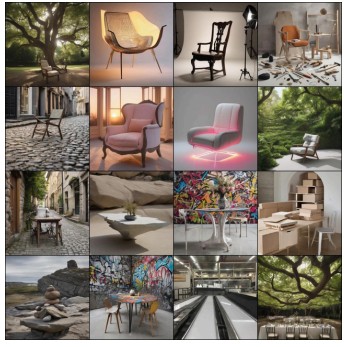

Model 2:

Samples from 2 object categories

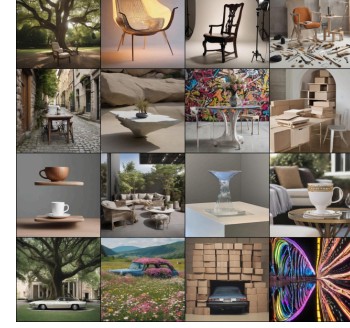

Model 4:

Samples from 4 object categories

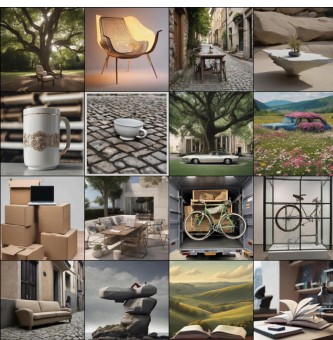

Model 8:

Samples from 8 object categories

Figure 18: Comparing Conditional-Vendi with Vendi on different fruit types generated by Stable Diffusion-XL.

# Type-specified object prompts

## Prompts:

A **Chair** is placed under a sprawling tree.
A **Sofa** is glowing in the light of a nearby fire.
A **Book** is balancing on the edge of a table.
A **Clock** is positioned in an office setup.
A **Lamp** is sitting under a hanging light bulb.
A **laptop** is half-hidden behind a stack of boxes.
A **car** is in the corner of a large warehouse.
The **cup** is precariously balanced on rocks.

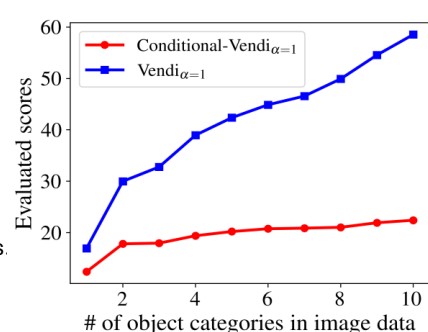

# Unspecified object prompts

## Prompts:

An **object** is placed under a sprawling tree.
An **object** is glowing in the light of a nearby fire.
An **object** is balancing on the edge of a table.
An **object** is positioned in an office setup.
An **object** is sitting under a hanging light bulb.
An **object** is half-hidden behind a stack of boxes
An **object** is in the corner of a large warehouse.
The **object** is precariously balanced on rocks.

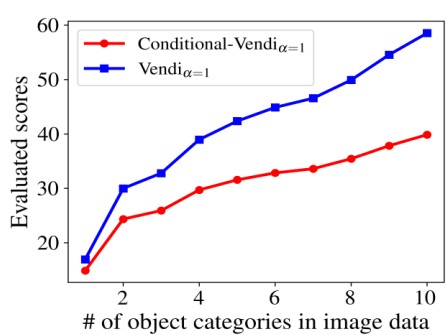

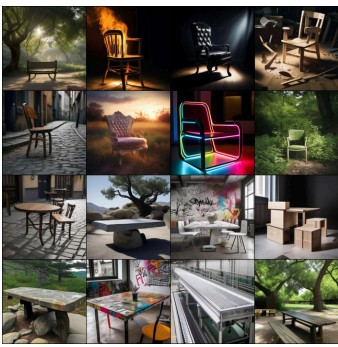

Model 2:

Samples from 2 object categories

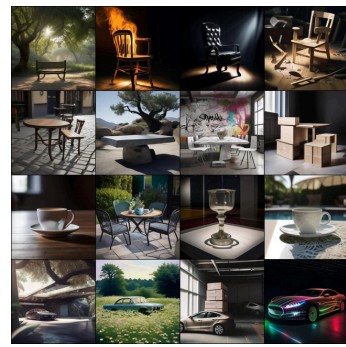

Model 4:

Samples from 4 object categories

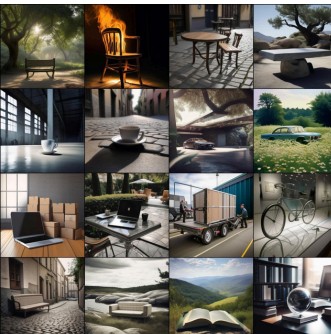

Model 8:

Samples from 8 object categories

Figure 19: Comparing Conditional-Vendi with Vendi on different object categories types generated by Kandinsky.

# Type-specified object prompts

## Prompts:

A **Chair** is placed under a sprawling tree.
A **Sofa** is glowing in the light of a nearby fire.
A **Book** is balancing on the edge of a table.
A **Clock** is positioned in an office setup.
A **Lamp** is sitting under a hanging light bulb.
A **laptop** is half-hidden behind a stack of boxes.
A **car** is in the corner of a large warehouse.
The **cup** is precariously balanced on rocks.

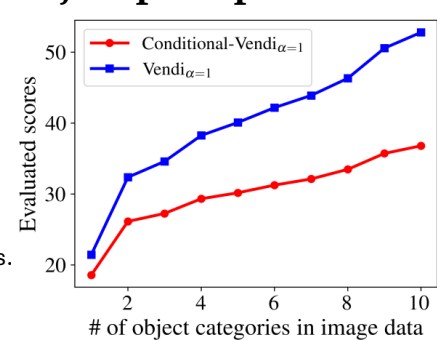

# Unspecified object prompts

## Prompts:

An **object** is placed under a sprawling tree.
An **object** is glowing in the light of a nearby fire.
An **object** is balancing on the edge of a table.
An **object** is positioned in an office setup.
An **object** is sitting under a hanging light bulb.
An **object** is half-hidden behind a stack of boxes.
An **object** is in the corner of a large warehouse.
The **object** is precariously balanced on rocks.

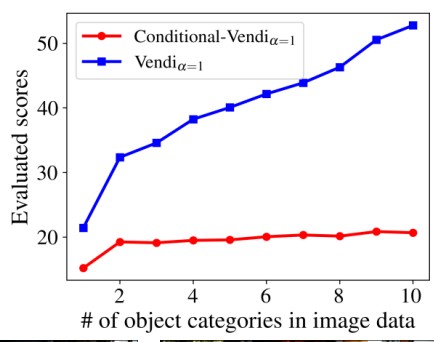

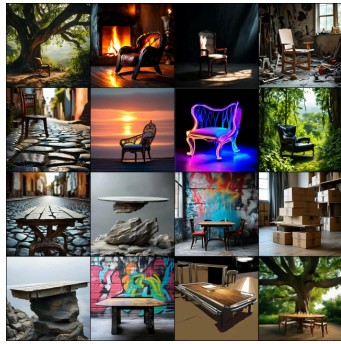
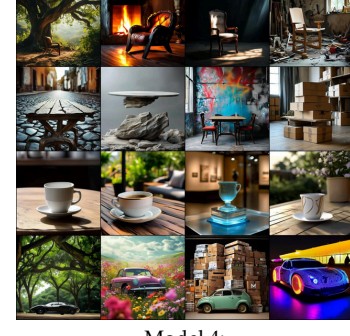
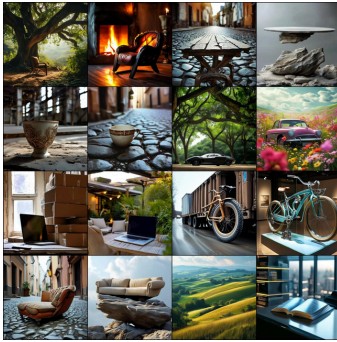

| Model 2: | Model 4: | Model 8: |
| Samples from 2 object categories | Samples from 4 object categories | Samples from 8 object categories |

Figure 20: Comparing Conditional-Vendi with Vendi on different object categories types generated by PixArt-$\Sigma$.

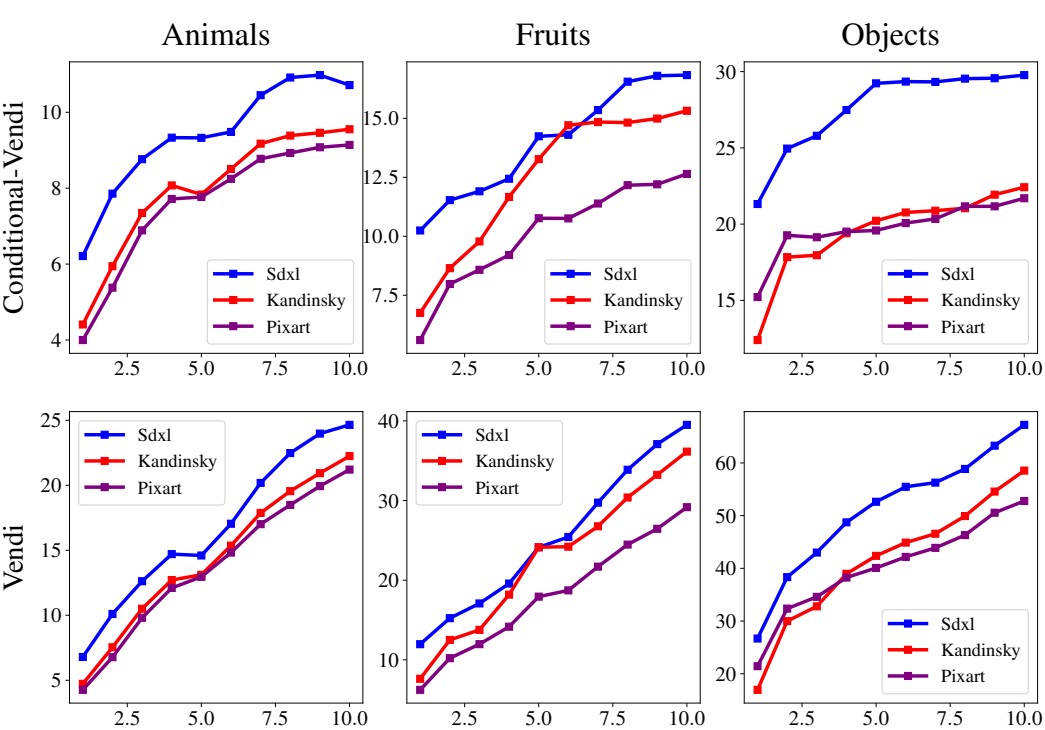

Figure 21: Evaluated (unconditional) Vendi and Conditional-Vendi scores of three text-to-image models in the category-based experiments with varying number of types within each of the categories: Animals, Fruits, Objects.

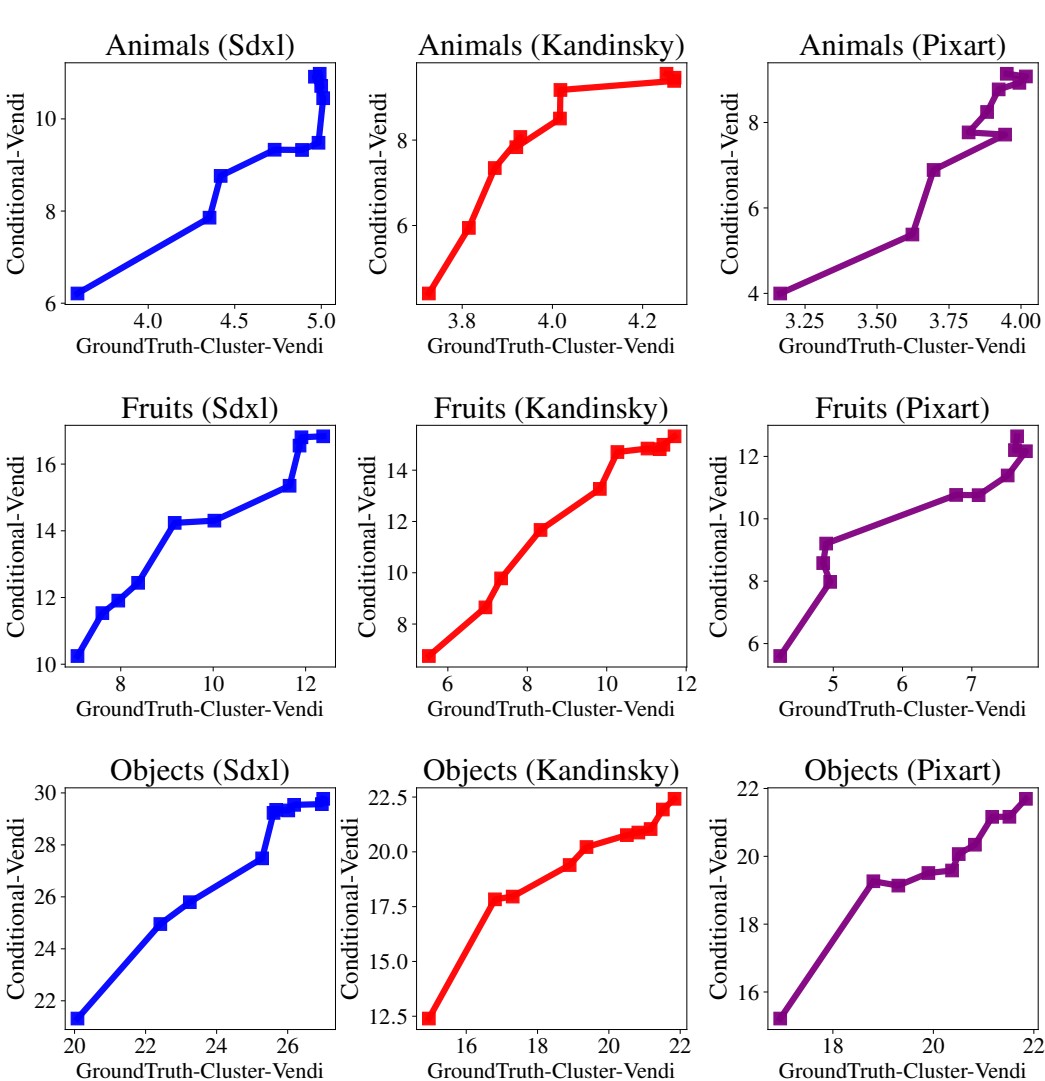

Figure 22: Comparing the Correlation of Conditional-Vendi with Groundtruth-Cluster-Vendi.

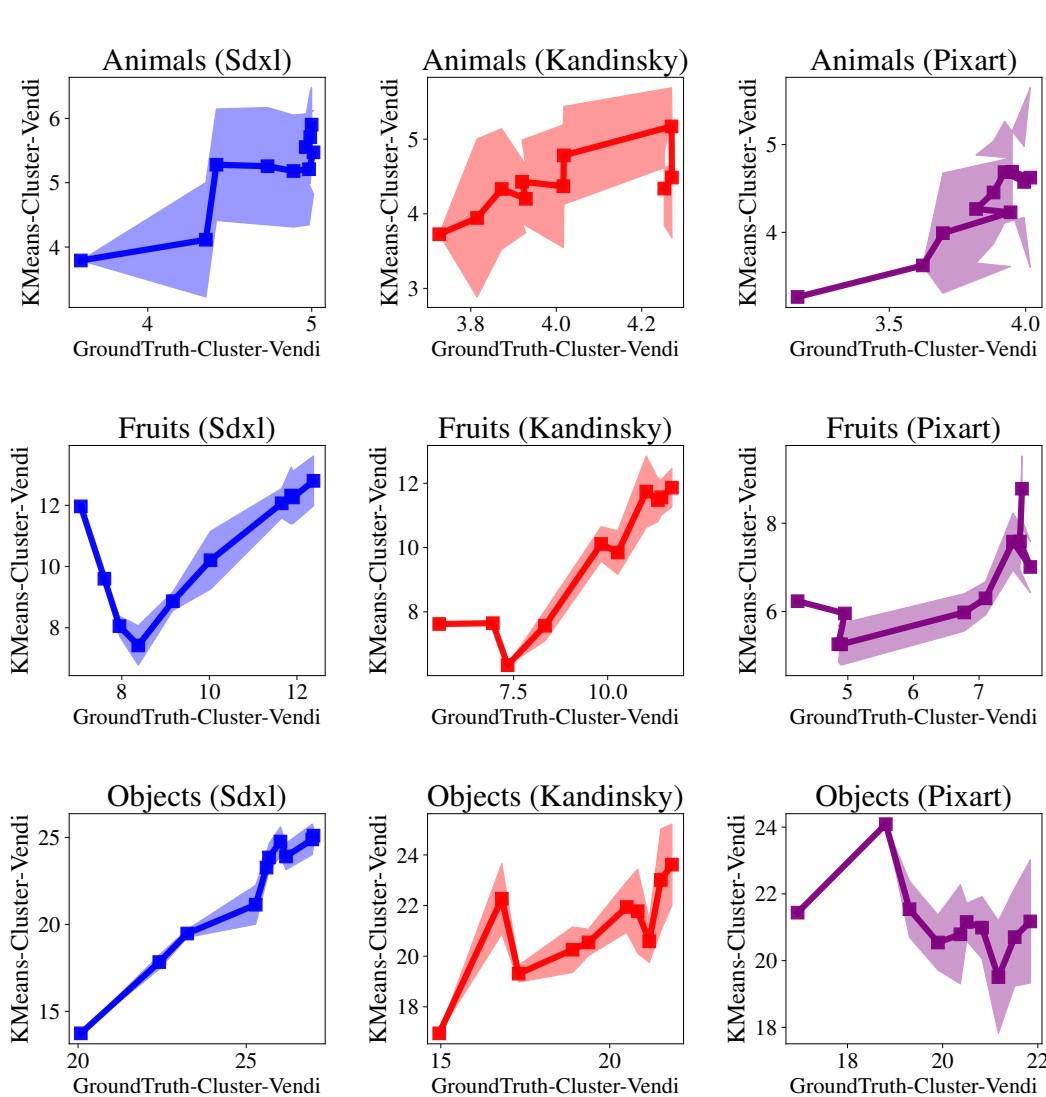

Figure 23: Comparing the Correlation of KMeans-Conditional-Vendi with Groundtruth-Cluster-Vendi.

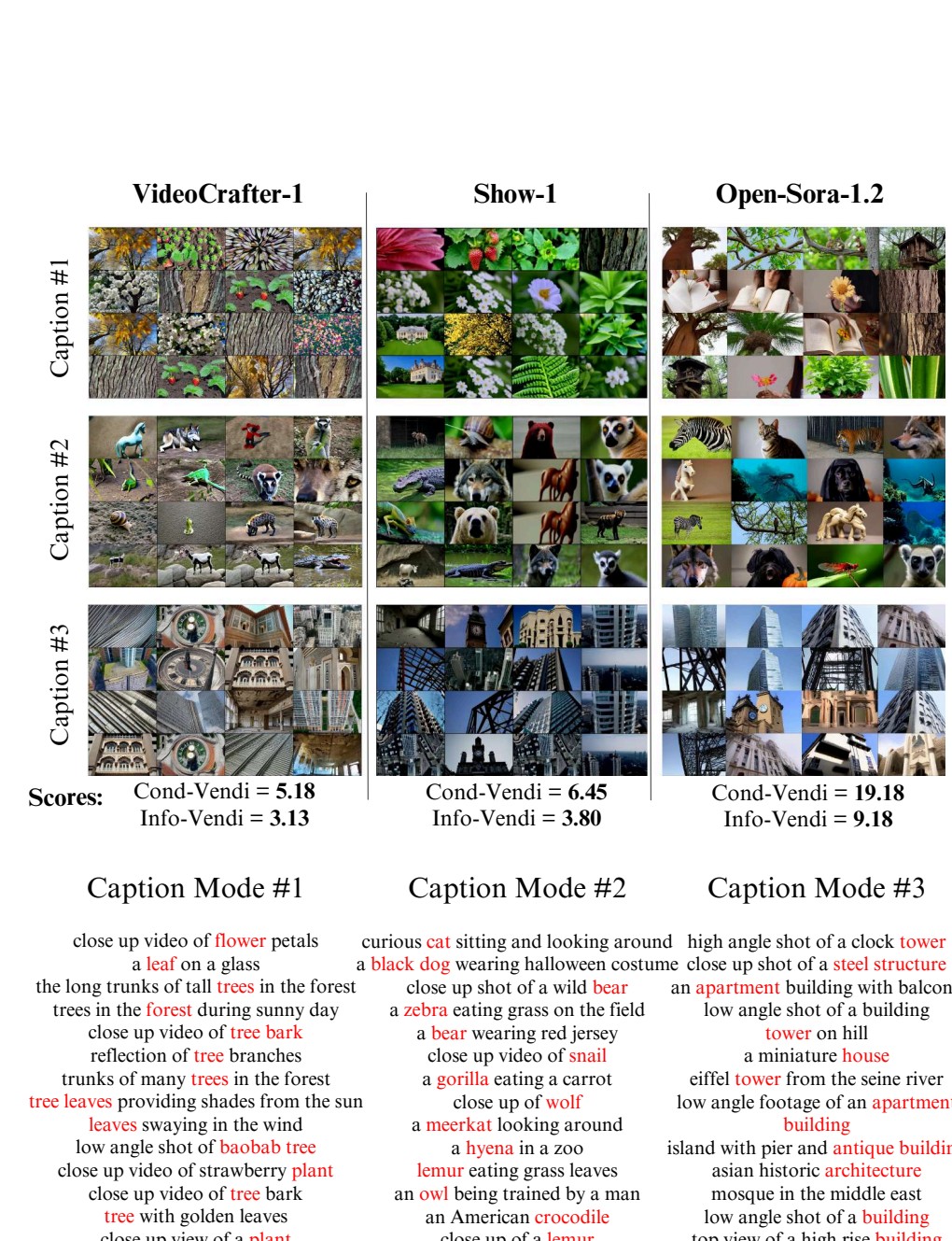

**Figure 24:** Measuring Conditional-Vendi and Information-Vendi for text-to-video models

---

**Algorithm 1** Conditional-Vendi and Information-Vendi

---

1: **Input:** Sample sets $\{\mathbf{x}_1, \ldots, \mathbf{x}_n\}$ and $\{\mathbf{t}_1, \ldots, \mathbf{t}_n\}$, Gaussian kernel bandwidths $\sigma_i^2, \sigma_t^2$, order $\alpha$.

2: **Compute kernel matrices**: $K_{\mathbf{X}} = \frac{1}{n}[k(\mathbf{x}_i, \mathbf{x}_j)]_{n \times n}$, $K_{\mathbf{T}} = \frac{1}{n}[k(\mathbf{t}_i, \mathbf{t}_j)]_{n \times n}$

3: **Perform eigendecomposition** on the $K_{\mathbf{X}}$, $K_{\mathbf{T}}$ and $\frac{1}{n}K_{\mathbf{X}} \odot K_{\mathbf{T}}$ matrices:

$$\{\lambda_1^{\mathbf{X}}, \ldots, \lambda_n^{\mathbf{X}}\} \leftarrow \text{Eigendecomposition}(K_{\mathbf{X}})$$

$$\{\lambda_1^{\mathbf{T}}, \ldots, \lambda_n^{\mathbf{T}}\} \leftarrow \text{Eigendecomposition}(K_{\mathbf{T}})$$

$$\{\lambda_1^{\mathbf{X},\mathbf{T}}, \ldots, \lambda_n^{\mathbf{X},\mathbf{T}}\} \leftarrow \text{Eigendecomposition}(\frac{1}{n}K_{\mathbf{X}} \odot K_{\mathbf{T}})$$

4: **Compute** $H_\alpha(\frac{1}{n}K_{\mathbf{X}})$, $H_\alpha(\frac{1}{n}K_{\mathbf{T}})$ **and** $H_\alpha(\frac{1}{n}K_{\mathbf{X}} \odot K_{\mathbf{T}})$ using their eigenvalues.

$$H_\alpha\left(\frac{1}{n}K_{\mathbf{X}}\right) \leftarrow \frac{1}{1-\alpha}\log\left(\sum_{i=1}^n (\lambda_i^{\mathbf{x}})^\alpha\right)$$

$$H_\alpha\left(\frac{1}{n}K_{\mathbf{T}}\right) \leftarrow \frac{1}{1-\alpha}\log\left(\sum_{i=1}^n (\lambda_i^{\mathbf{T}})^\alpha\right)$$

$$H_\alpha\left(\frac{1}{n}K_{\mathbf{X}} \odot K_{\mathbf{T}}\right) \leftarrow \frac{1}{1-\alpha}\log\left(\sum_{i=1}^n (\lambda_i^{\mathbf{X},\mathbf{T}})^\alpha\right)$$

5: **Compute Conditional-Vendi and Information-Vendi**

$$\text{Conditional-Vendi}_\alpha\left(x_1, \ldots, x_n | t_1, \ldots, t_n\right) \leftarrow \exp\left(H_\alpha\left(\frac{1}{n}K_{\mathbf{X}} \odot K_{\mathbf{T}}\right) - H_\alpha\left(\frac{1}{n}K_{\mathbf{T}}\right)\right)$$

$$\text{Information-Vendi}_\alpha\left(x_1, ., x_n; t_1, ., t_n\right) \leftarrow \exp\left(H_\alpha\left(\frac{1}{n}K_{\mathbf{X}}\right) + H_\alpha\left(\frac{1}{n}K_{\mathbf{T}}\right) - H_\alpha\left(\frac{1}{n}K_{\mathbf{X}} \odot K_{\mathbf{T}}\right)\right)$$

6: **Output:** Conditional-Vendi and Information-Vendi

---

| Kandinsky | SDXL | GigaGAN | FLUX |

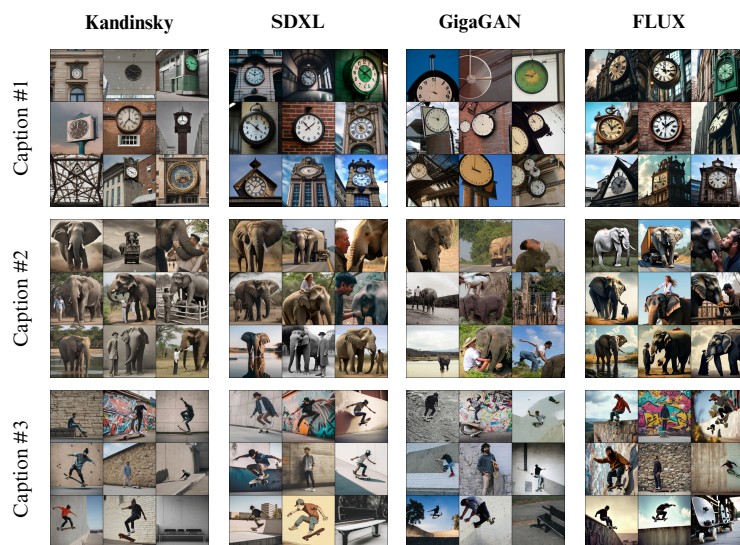

**Caption Mode #1**

A building displaying a clock showing the time to be 6 oclock.
A clock hanging from the ceiling of a building.
A large metal green clock hanging from the side of a building.
A clock that is on top of a sign.
A large clock mounted to a brick wall.
A large clock hanging off the side of a tall building.
A clock in near the triangular roof of a large building.
A large clock and a sign on top of a building.
A large clock mounted to the side of a building.

**Caption Mode #2**

The elephant has a large white spot on its abdomen.
The truck driver hauls an elephant down the highway.
A man getting a kiss on the neck from an elephant's trunk
A large elephant walking next to a man
A woman in white shirt climbing onto an elephant.
A man is leaning over a fence offering food to an elephant/
A large elephant standing on the side of a lake.
A man standing next to an elephant who stole his hat with it's trunk.
A man standing near an elephant with its trunk outstretched.

**Caption Mode #3**

A young man riding a skateboard on a stone wall.
A man balancing on a skateboard in front of a graffiti covered wall.
A man doing a trick on a wall with a skateboard.
Bearded skateboarder maintains balance while skating up wall.
A man standing next to a stone wall while holding a skateboard.
There is a man skateboard on the side of a wall.
a guy skate boarding on the edge of a wall
A man on a skateboard is trying to jump over a wall.
two black and white skate boards under a black steel bench

Figure 25: Effect of text kernel bandwidth on Conditional-Vendi and Information-Vendi scores

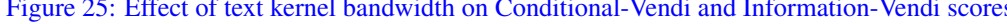
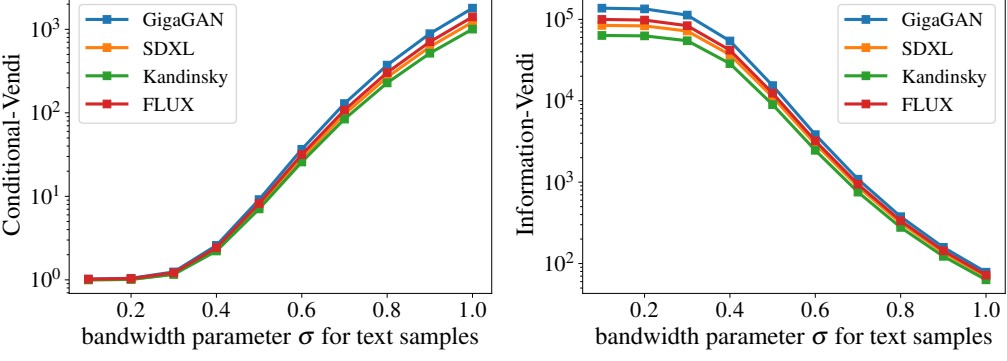

Figure 26: Effect of text kernel bandwidth on Conditional-Vendi and Information-Vendi scores

