# OpenReview forum: "An Information-Theoretic Approach to Diversity Evaluation of Prompt-based Generative Models"
_ICLR.cc/2025/Conference — Submitted to ICLR 2025_

### Official Review · Reviewer_f3gh · 2024-10-31

**Soundness:** 3
**Presentation:** 2
**Contribution:** 1
**Rating:** 3
**Confidence:** 3

**Summary:**

This paper focuses on distinguishing different sources for diversity evaluation of prompt-based generative models, i.e. prompt-induced diversity and model-induced diversity. Specifically, this paper proposed a new diversity evaluation metric called *condtional-Vendi* score, which is an extension of *Vendi*, but focuses on model-induced diversity rather than overall diversity. Combining with another proposed *information-Vendi* score, the *Vendi* score can be recovered by a multiplication. Experiments on image and text generation tasks show that condtional-Vendi is correlated with model-induced diversity.

**Strengths:**

* The idea is sound to separate different sources of diversity, i.e., prompt-induced and model-induced.
* The proposed diversity metric is reasonable for model-induced diversity, and have the advantage that do not need repeated sampling from a single prompt.

**Weaknesses:**

The main weakness of this paper is its contribution. The proposed condtional-Vendi and information-Vendi score are just exponentials of the conditional entropy and mutual information defined by Sanchez Giraldo et al. (line 241, 246), and the calculation method seems to directly inherit from Vendi proposed by Friedman & Dieng (line 317). The combination is straightforward and I did not find any new challenges that is addressed by this paper. As a result, I view this paper as just an incremental work, and not enough for acceptance.

**Questions:**

1. What's new of your conditional-Vendi compared with the conditional entropy proposed by Sanchez Giraldo et al., besides the exponential?
2. Follow which equation can the conditional-Vendi score be calculated? It's better to write an explicit algorithm, perhaps in the appendix.
3. What do you want to convey by Section 5 (the statistical interpretations)? Is it discussing the implementations, or just broader discussions? A better leading sentence may be necessary in the beginning.

---

> ### Author Response · Authors · 2024-11-25
>
> We thank Reviewer f3gh for his/her thoughtful feedback on our work. We are glad to hear that the reviewer finds the idea proposed in our work “sound”. Regarding the reviewer’s comments and questions:
>
> **1-”What's new of your conditional-Vendi compared with the conditional entropy proposed by Sanchez Giraldo et al., besides the exponential?”**
>
> We believe that our technical contribution extends beyond merely proposing the application of the matrix-based conditional entropy for evaluating model-induced diversity in conditional generative models. Specifically, we provide an interpretation of the conditional entropy in Theorem 1, which establishes a connection between the conditional entropy defined by Sanchez Giraldo et al. in [1] and the average of the (unconditional) entropy over the components of a mixture text distribution $P_T$. To the best of our knowledge, this theoretical result is novel and has not been previously addressed in the literature.
>
> For more details, please refer to Item 1 in our general response.
>
> **2- “Follow which equation can the conditional-Vendi score be calculated?"**
>
> The equation in Line 263 combined with entropy definition in Equation (3) gives the formula for computing the Conditional-Vendi score.
>
> **3- "It's better to write an explicit algorithm, perhaps in the appendix.”**
>
> Following the reviewer’s suggestion, we have added Algorithm 1 to the revised Appendix B.6. We thank Reviewer f3gh for the suggestion.
>
> **4-”What do you want to convey by Section 5 (the statistical interpretations)?”**
>
> In Section 5, we first derive the statistic estimated by the matrix-based conditional entropy function defined in [1]. Note that the conditional entropy definition $H(x_1,\ldots , x_n|t_1,\ldots ,t_n)$ in [1] is a complex function of the samples, and it is not clear what target statistic this score aims to estimate as the sample size $n$ approaches infinity. In Proposition 1 and Corollary 1, we derive the target statistic to reveal what function of the joint (text,image) distribution $P_{T,X}$ is estimated by the matrix-based conditional entropy defined in [1].
>
> In Theorem 1, our goal is to interpret the matrix-based conditional entropy $H(X|T)$ defined by Sanchez Giraldo et al. Please note that the **matrix-based** conditional entropy measure $H(X|T)$ defined in [1] is **not equal** to the average of instance-specific entropies $H(X|T=t)$ over $t \sim P_T$, which means
>
> $$H(X|T)  \text{   }  {\large\neq} \text{   } \mathbb{E}_{t \sim P_T}[H(X|T=t)]$$
>
> Theorem 1 addresses this gap and interprets the matrix-based conditional entropy $H(X|T)$ as averaging the entropies $H(X|C=i)$ for the components of a mixture text distribution $P_T$. These components are defined based on the bandwidth parameter $\sigma$ of the text Gaussian kernel function, which determines the threshold for identifying two embedded text vectors as “close” or “far.” For more details, please refer to Item 1 in our general response.
>
> [1] Sanchez Giraldo et al., “Measures of entropy from data using infinitely divisible kernels”. IEEE Transactions on Information Theory, 2014

---

> > ### Comment · Reviewer_f3gh · 2024-11-28
> >
> > Thank you for your response. I have read the updated paper, please use the updated line numbers in the following discussions.
> >
> > **Regarding the contribution:**
> >
> > I notice that you have added the theoretical analysis as an important contribution. Now I understand the role of Section 5 after reading your explanations. However, I still have two main concerns:
> >
> > * As stated in your abstract, the main contribution of this paper seems to be the decomposition of the total entropy into 2 parts, rather than bringing up with new theoretical analysis. Under this view, it is the first priority to validate this contribution. However, I find this point has already been addressed by Giraldo et al., as stated in line 247-256. Please explain the novelty of your work on this point (the "exponential" does not count).
> >
> > * As for your theoretical contribution on $H(X|T)\approx \mathbb{E}_I\ H(X|G=I)$, I acknowledge its novelty, but the contribution is relatively weak. In my understanding (based on your statement in line 339-341), Thm. 1 is mainly a support for experiment design, providing an approach for utility analysis of proposed metrics. Therefore, this contribution cannot stand out by itself, without the novelty of the proposed metrics.
> >
> > **Minor suggestions:**
> >
> > * I suggest to use "Renyi entropy" v.s. "Shannon entropy" to better distinguish your notations, rather than "classical entropy".
> > * Symbols in line 1818-1821 do not align well with others in Algorithm 1. Please revise if possible.

---

> > > ### Author Response · Authors · 2024-11-28
> > >
> > > We thank Reviewer f3gh for the thoughtful feedback on our response. Regarding the points raised in the feedback:
> > >
> > > **1- Contribution summarization in the abstract**
> > >
> > > We would like to clarify that our work’s main contribution is to highlight the task of *model-induced* and *prompt-induced* diversity evaluation of prompt-based generative models, and to propose the first metrics to quantify these diversity terms.  Specifically, we propose the application of matrix-based information measures to address this task. We theoretically and numerically interpret this evaluation process as the aggregation of (unconditional) entropy measures within each mode of a multi-modal text distribution. The proposed approach nicely connects to the existing Vendi score for diversity evaluation of unconditional generative models, and reveals that the Vendi score can be decomposed into the multiplication of two terms quantifying the model-induced and prompt-induced diversity.
> > >
> > > To address the reviewer’s concern, we have revised the abstract to clearly state our work’s contributions.
> > >
> > > **2- The technical contribution in Theorem 1**
> > >
> > > We would like to emphasize the theoretical contribution in the proof of Theorem 1. We believe this proof is highly non-trivial, as it involves analyzing the Kronecker product of the Gaussian kernel covariance matrix for the text variable $T$  and the kernel covariance matrix for the generated data variable  $X$ . The use of the Kronecker product is motivated by Proposition 1, which demonstrates the equivalence between the Hadamard product of kernel matrices and the Kronecker product of kernel covariance matrices.
> > >
> > > We believe that the proof techniques can be more generally useful in other applications of matrix-based information measures and quantum information theory. To the best of our knowledge, these analytical tools and results have not been presented in the existing literature, including the work of Giraldo et al. (2014). For further details, we kindly refer the reviewer to Appendix A, where the complete proofs are provided.
> > >
> > > **3- Minor revisions**
> > >
> > > Thank you for suggesting the edits, we have made the recommended modifications to the revised draft.

---

> > > > ### Comment · Reviewer_f3gh · 2024-11-29
> > > >
> > > > Thank you for your response. However, it seems that you fail to understand my concerns correctly.
> > > >
> > > > **Regarding the main contribution:**
> > > >
> > > > You wrote "to propose the *first* metrics to quantify these diversity terms" in your response. However in my view, Giraldo et al. (2014) have proposed similar metrics, which already achieves the goal that separating model-induced and prompt-induced diversity. Why do you claim that your proposed metrics are the *first*?
> > > >
> > > > **Regarding the contribution of Thm. 1:**
> > > >
> > > > As said in my previous response, I do not question about the novelty of Thm. 1 itself. What I mean is, do not avoid discussions regarding the novelty of your metrics, by claiming that your theorem is also a sufficient contribution (it is far from being *sufficient* itself). So please focus on my first concern.

---

> > > > > ### Author Response · Authors · 2024-12-03
> > > > >
> > > > > Thank you for your response. To be clear, when we said “the first metric to quantify these diversity terms”, we meant no previous works have studied the task of quantifying “model-induced and prompt-induced diversity”. Entropy is a general uncertainty measure, where uncertainty can broadly refer to any form of randomness in the system. For example, in the communication problem, one can think of the entropy of the noise in the channel, which is not interpreted as diversity.
> > > > >
> > > > > The work by Grialdo et al, 2014 does not study the quantification of diversity, and their introduced information measures are not applied to generative models. In our work, we propose the application of conditional entropy to address a new task which has not been studied before. The updated abstract reflects this point that we focus on a new diversity quantification task that has not been studied before and apply information measures to address the task.

---

### Official Review · Reviewer_Uq1Z · 2024-11-02

**Soundness:** 3
**Presentation:** 3
**Contribution:** 2
**Rating:** 3
**Confidence:** 4

**Summary:**

This paper focuses on the diversity in prompt-based generative models. Specifically, the authors want to quantify the prompt-induced and model-induced diversity in samples separately. An information-theoretic approach for internal diversity quantification based on kernel-based conditional entropy of the generated data is discussed, where the Conditional-Vendi score and the Information-Vendi score are applied to measure the statistical relevance between the generated data and text prompts.

**Strengths:**

Originality: the problem of diversity quantification is indeed interesting and useful in generative models.

Clarity: the paper is well-organized and straightforward to follow, with supportive numerical results and clearly stated details.

Overall the paper is a nice reading.

**Weaknesses:**

My major concern is that the contribution of this work is incremental. As stated by the paper in Section 3.2, the conditional entropy $H_\alpha(X|T)$ is already developed by Giraldo et al (2014), while the Conditional-Vendi score proposed by the authors is $\exp\left(H_\alpha(X|T)\right)$, which is essentially the same. In addition, existing literature has considered Renyi entropy of the probability model as the model's unconditional diversity score, and the idea of constructing conditional diversity score based on these results does not compose of sufficient contribution. Thus, I feel the contribution is incremental.

Another concern I have is the potential applications of the proposed diversity measure. In most scenarios mentioned by the paper, say statistical interpretation in Section 5 and some motivating examples, prompts can be easily clustered, so what about just calculate the unconditional entropy for each cluster of prompts, and then use some weighted average to combine them? I personally feel this is a naive but reasonable way to measure the conditional diversity of the model and should serve as a baseline. If prompts cannot be easily clustered and come from some complex distributions, then mathematically the conditional entropy would depend heavily on the distribution of the prompt distribution, the the concrete meaning of this measure is unclear to me.

For numerical results, though they are clear to me, after reading it I can still not tell if conditional entropy is a good measure, as there is no baseline compared (which is reasonable as this is a relatively new problem, but the authors should at least consider some naive methods such as the one mentioned above, to see if there are some scenarios that their proposed method works while naive methods don't) and the meaning of values reported (say, figure 5-7) remains unclear to me. Why can these figures and values justify that the conditional entropy is a good measure of model's diversity?

**Questions:**

(Contribution) As mentioned in Weakness, could the authors explain their contributions to prior works such as Giraldo et al (2014)?

 (Applications) For prompts that can be easily clustered, why don't we just average unconditional entropy for each cluster? For more general distribution, what's the practical meaning of the conditional entropy when the distribution $\mathcal{T}$ changes?
 (Numerical) Why can current numerical results illustrate the advantages of proposed metric?

---

> ### Author Response · Authors · 2024-11-25
> **Authors' Response (Part 1)**
>
> We thank Reviewer Uq1Z for his/her thoughtful feedback on our work. We are glad to hear that the reviewer finds the problem studied in our work “interesting and useful ”. Regarding the reviewer’s comments and questions:
>
> **1- Technical Contribution of our work**
>
> We believe that our technical contribution extends beyond merely proposing the application of the matrix-based conditional entropy for evaluating model-induced diversity in conditional generative models. Specifically, we provide an interpretation of the conditional entropy in Theorem 1, which establishes a connection between the conditional entropy defined by Sanchez Giraldo et al. in [1] and the average of the (unconditional) entropy over the components of a mixture text distribution $P_T$. To the best of our knowledge, this theoretical result is novel and has not been previously addressed in the literature. For further details, please refer to Item 1 in our general response.
>
> **2- “the idea of constructing conditional diversity score based on these results does not compose of sufficient contribution”**
>
> Please note that the **matrix-based** conditional entropy measure $H(X|T)$ defined in [1] is not equal to the average of instance-specific entropies $H(X|T=t)$ over $t \sim P_T$, i.e.,
>
> $$H(X|T) \neq \mathbb{E}_{t \sim P_T}[H(X|T=t)]$$
>
> As a result, our proposal for model-induced diversity quantification is not simply the averaging of entropies $H(X|T=t)$ for data generated for each prompt $t \sim P_T$. In Theorem 1, we interpret this process as averaging the entropies $H(X|C=i)$ for the components of a mixture text distribution $P_T$. These components are defined based on the bandwidth parameter $\sigma$ of the text Gaussian kernel function, which determines the threshold for identifying two embedded text vectors as “close” or “far.”
>
>
> **3- “so what about just calculate the unconditional entropy for each cluster of prompts, and then use some weighted average to combine them?”**
>
> We implemented the reviewer’s suggestion to define a golden metric, “GroundTruth-Cluster-Vendi,” which computes the average Vendi scores over text clusters assumed to be provided by an oracle. Please refer to Item 2 in our general response for the numerical results showing the correlation between the GroundTruth-Cluster-Vendi and our defined Conditional-Vendi scores.
>
> While the intuition behind the Cluster-Vendi score suggested by the reviewer is meaningful, note that, in a general evaluation task, we **do not have access** to the ground truth clusters. Consequently, computing sample clusters to evaluate the Cluster-Vendi score becomes necessary. This task can be intractable in general cases, as we may not know the number of clusters or whether the samples are well-clustered. Even if the number of clusters is known, the standard K-Means algorithm can converge to different solutions depending on its initialization, as the K-Means optimization problem is highly non-convex. In our numerical experiments, we also observed that the KMeans-Cluster-Vendi score exhibits significant variance due to the random initialization of K-Means clustering. In contrast, our defined Conditional-Vendi score can be computed deterministically through the eigendecomposition of kernel matrices, which can be computed with no error using the solution to the tractable eigendecomposition task.
>
> **4- Measuring the conditional diversity using a clustered Vendi score**
>
> We agree with the reviewer that the GroundTruth-Cluster-Vendi can serve as a valuable golden metric for evaluating the Conditional-Vendi score. For further details on this suggestion, please refer to Item 2 in our general response. We thank the reviewer for this great suggestion.
>
> **5- "If prompts cannot be easily clustered and come from some complex distributions… then the concrete meaning of this measure is unclear to me."**
>
> In a general scenario where the text samples may not be well-clustered, the Conditional-Vendi score relies on the eigenvectors of the kernel covariance matrix to define **soft clusters** over the data. Note that, given a kernel function $k$, the kernel covariance matrix is a positive semi-definite (PSD) matrix with orthogonal eigenvectors. In our proposed diversity evaluation, each eigenvector of the kernel covariance matrix represents a **soft cluster** of the data.
>
> If the data distribution $P_T$ consists of well-separated components, then with high probability, the drawn samples will form well-defined clusters, and the soft clusters derived from the eigenvectors will closely resemble hard clusters. However, even when the samples are not well-clustered, the interpretation of soft clusters based on the eigenvectors of the kernel covariance matrix remains applicable to the proposed diversity calculation.
>
> [1] Sanchez Giraldo et al., “Measures of entropy from data using infinitely divisible kernels”. IEEE Transactions on Information Theory, 2014

---

> > ### Author Response · Authors · 2024-11-25
> > **Authors' Response (Part 2)**
> >
> > **6- “the authors should at least consider some naive methods such as the one mentioned above”**
> >
> > We used the GroundTruth-Cluster-Vendi score as a golden metric in the experiment settings of Figures 12-20 where we know the underlying clusters. We observed that the Conditional-Vendi score correlated well with the GroundTruth-Cluster-Vendi score. For more details, please refer to Items 2 and 3 of the general response.
> >
> >
> > **7-”could the authors explain their contributions to prior works such as Giraldo et al (2014)?”**
> >
> > Please refer to Item 1 of our general response.
> >
> > **8- “For prompts that can be easily clustered, why don't we just average unconditional entropy for each cluster?”**
> >
> > Please refer to Items 3 and 4 of Part 1 of this response.
> >
> > **9- “Why can current numerical results illustrate the advantages of proposed metric?”**
> >
> > Please refer to Items 2 and 3 of the general response.

---

### Official Review · Reviewer_awod · 2024-11-02

**Soundness:** 2
**Presentation:** 3
**Contribution:** 1
**Rating:** 3
**Confidence:** 5

**Summary:**

The paper introduces an information-theoretic approach to evaluate the diversity of prompt-based generative models, focusing on decomposing the diversity into prompt-induced and model-induced components. The authors achieve this by decomposing the Vendi score into the Conditional-Vendi score, which measures internal diversity not related to prompt variations, and the Information-Vendi score, which assesses the statistical correlation between generated outputs and their corresponding prompts. The authors validate their methodology through various experiments on text-to-image, text-to-video, and image-captioning models.

**Strengths:**

- The motivation for decomposing the diversity of model generation is clear, despite that a more fine-grained structural analysis would be preferred.
- The theoretical interpretation of the scores provides sufficient insight for the decomposition.
- The paper is well-organized and logically fluent.

**Weaknesses:**

While this paper tackles an important problem, the following weaknesses prevent it from being accepted.

- First of all, I think the paper lacks novelty in terms of the proposed method. The concept of conditional entropy $H_\alpha$ is nothing new, and the proposed conditional-Vendi score is simply its exponential version. If the paper is not contributing through new concepts or definitions, it should at least demonstrate some (new) important properties or use cases. Unfortunately, the only property discussed in the paper is its relevant stability when more diverse prompts are used, which, theoretically speaking, is expected. In such a case, the only novelty I can admit is that the paper demonstrates in experiments that the conditional entropy behaves normally in generative models. The authors are expected to provide a more comprehensive analysis of the given entropy and demonstrate a wider range of use cases if the concept itself is inherited from existing literature.
- Even if we consider only the stability property, the range of experiments in the paper is still limited. All experiments are simply measuring the scores with varying amounts of input clusters (groups), and the way that the authors make prompts diverse is also rigid. It is entirely unclear, based on existing experiments, that the proposed score can serve as an effective diversity measurement across a wide range of text prompts types. What's more, the authors do not show any support that the measure is accurate, except for this extremely intuitive "increasing curve". No golden metric, or even any proxy metric, has been introduced for comparison. This means that the "diversity measurement" claim is not convincing in the first place.
- Theoretically speaking, a large portion of the discussion about the selection of kernels is missing. From my understanding, since the authors only measure diversity from the generated data (rather than the underlying softmax distribution), the similarity metric used for text and images is very important and can largely change the metric. It is, therefore, necessary for the authors to carefully analyze the choice of kernels and make a more well-studied claim w.r.t. the proposed metric. However, notice that addressing this bullet is insufficient for making the paper reach the acceptance bar unless the first two weaknesses can be addressed.

**Questions:**

No additional question other than the major weakness above.

---

> ### Author Response · Authors · 2024-11-25
> **Authors' Response (Part 1)**
>
> We thank Reviewer awod for his/her thoughtful feedback on our work. We are glad to hear that the reviewer finds the work “well-organized and logically fluent”. Regarding the reviewer’s comments and questions:
>
> **1- “it should at least demonstrate some (new) important properties or use cases”**
>
> We believe that we have demonstrated novel and important properties of the matrix-based conditional entropy as defined by Sanchez Giraldo et al. in [1]. It is important to note that the **matrix-based** entropy measure $H(X|T)$ defined in [1] lacks the standard interpretation of Shannon conditional entropy, which corresponds to the averaged instance-specific entropy $H(X|T=t)$ over $t \sim P_T$. As a result, the relationship between $H(X|T)$ as defined in [1] and the average of instance-specific entropies $H(X|T=t)$ remains unclear.
>
> Our main technical contribution, Theorem 1, aims to address this gap and provides an interpretation of the matrix-based conditional entropy $H(X|T)$. We believe this contribution offers useful insights into the properties of matrix-based conditional entropy measures. For more details, please refer to Item 1 in our general response.
>
> **2-“In such a case, the only novelty I can admit is that the paper demonstrates in experiments that the conditional entropy behaves normally in generative models.”**
>
> We would like to emphasize the theoretical results presented in Section 5, which are, to the best of our knowledge, entirely novel. Theorem 1 provides an operational interpretation for the conditional entropy $H(X|T)$ defined by [1]. As far as we know, this has not been addressed in the existing literature. For more details, please refer to Item 1 in our general response.
>
>
> **3- “No golden metric, or even any proxy metric, has been introduced for comparison.”**
>
> As discussed in Item 2 of our general response, to the best of our knowledge, our work is the first to explore the diversity decomposition of conditional generative models. Consequently, there are no established metrics in the literature to separately quantify model-induced and prompt-induced diversity of generated samples that we could use as a golden metric.
>
> In the revision, we have followed the suggestion of Reviewer Uq1Z to define and utilize the GroundTruth-Cluster-Vendi metric as a surrogate golden metric. The GroundTruth-Cluster-Vendi assumes knowledge of sample clusters and computes the average (unconditional) Vendi score of samples within each cluster. In the nine simulated settings where we had access to the ground truth sample clusters, the GroundTruth-Cluster-Vendi score exhibited strong correlation with the conditional-Vendi score. For additional details, we kindly refer the reviewer to Item 2 of the general response.
>
> **4- “the range of experiments in the paper is still limited”**
>
> In the revision, we conducted the diversity evaluation for 9 additional settings, combining three categories with three SOTA text-to-image models. For more details, please refer to Item 3 of the general response.
>
> **5- “All experiments are simply measuring the scores with varying amounts of input clusters (groups)”**
>
> As discussed in Item 3, to the best of our knowledge, there are no baseline metrics in the literature to quantify model-induced diversity. Consequently, we could not assess the Conditional-Vendi score by measuring its correlation with an existing baseline metric. In the original submission, we simulated numerical settings where the ground truth ranking of model-induced diversity across several models was known, and demonstrated that the Conditional-Vendi score accurately reflected this ground truth ranking.
>
> In the revision, we also included a numerical comparison with the newly defined golden metric, GroundTruth-Conditional-Vendi. However, it is important to note that computing the GroundTruth-Conditional-Vendi score requires knowledge of the underlying clusters. For a general dataset, this computation becomes intractable due to the non-convex nature of K-Means and other standard clustering optimization problems.
>
> [1] Sanchez Giraldo et al., “Measures of entropy from data using infinitely divisible kernels”. IEEE Transactions on Information Theory, 2014

---

> > ### Author Response · Authors · 2024-11-25
> > **Authors' Response (Part 2)**
> >
> > **6- “Theoretically speaking, a large portion of the discussion about the selection of kernels is missing.”**
> >
> > We would like to clarify that Theorem 1 in the text suggests the use of a Gaussian kernel for text prompts. The theorem explains the role of the Gaussian kernel bandwidth $\sigma$ in characterizing text clusters (here, referring to soft clusters that assign probabilities to each sample belonging to clusters). Consequently, we selected the text kernel to be Gaussian, with a properly tuned bandwidth $\sigma$ that determines whether text prompts are considered “close” or “far” in alignment with human perception.
> >
> > For the generated image kernel, we follow the kernel choice used in the unconditional Vendi score, as discussed in detail in related works on Vendi diversity calculation. Specifically, we followed the numerical experiments in [2-4] for selecting the Gaussian kernel bandwidth for both the generated image and video data.
> >
> > To address the reviewer's comment on the choice of kernel function in the numerical evaluation, we performed the diversity quantification for MS-COCO text data as input to three text-to-image models using different bandwidth parameter $\sigma$ for the Gaussian kernel of text samples. The numerical results can be found in Figure 26 in the revised Appendix B.8, which show consistent rankings of the three models under different bandwidth parameters.
> >
> >
> > [2] Friedman, D. & Dieng, A. B. “The Vendi Score: A Diversity Evaluation Metric for Machine Learning”, TMLR 2023\
> > [3] Ospanov, et al. “Towards a Scalable Reference-Free Evaluation of Generative Models” , NuerIPS 2024\
> > [4] Amey P. Pasarkar, Adji Bousso Dieng “Cousins Of The Vendi Score: A Family Of Similarity-Based Diversity Metrics For Science And Machine Learning”, AISTATS 2024

---

### Official Review · Reviewer_3Epu · 2024-11-05

**Soundness:** 4
**Presentation:** 4
**Contribution:** 3
**Rating:** 8
**Confidence:** 3

**Summary:**

This paper proposes a metric for evaluating the diversity of generative models. Unlike previous approaches which focus on unconditional generations, this paper uses an information-theoretic decomposition of the entropy into conditional entropy (measuring variation in generation when conditioned on a prompt) and mutual information (measuring the remaining variation not explained by the conditional entropy). The first term is expressed as a conditional Vendi score and the second as an information Vendi score. It is shown that the Vendi score is the product of these two, and so generative models with similar Vendi scores can be compared based on their conditional diversity given a prompt. Theoretical justfication for the proposed metrics is provided, and it is also shown how the metric can be estimated from finite samples. The proposed metrics are explored in different contexts, including text-to-image, text-to-video, and image-captioning models. The results correspond with the intuitive interpretations of the scores.

**Strengths:**

Strengths of this paper include:
- The proposed metrics occupy a clearly defined niche in the literature, providing a rich and informative way to evaluate prompt-based generative models. This is useful for better comparisons of generative models (even applicable across different modalities) and for better model development by elucidating the shortcomings of current models.
- Solid theoretical justification and empirical evidence are provided to support the proposed metrics.
- The paper is exceedingly well-written. Prose is clear and mathematical statements are presented in a precise yet intuitive way.

**Weaknesses:**

Weaknesses of this paper include:
- The proposed approach can be viewed as incremental, since it is the combination of two ideas: the Vendi score and matrix-based conditional entropy. However, the execution of the paper is very good and is targeted at a gap in the current literature, so I do not count this as a major weakness.
- The contextualization and interpretation of Theorem 1 could be improved.
- Qualitative results could be improved in some places. For example, showing some of the generations from the models in Figure 4 would help to better clarify the tradeoffs of the models in terms of conditional Vendi vs. information Vendi score.

**Questions:**

My opinion of the work is already high so I do not have many questions. However, in light of Figure 6, it seems that it would be possible to achieve a high conditional Vendi score by a model that has poor alignment with the prompt. Is there any way that the proposed conditional/information Vendi decomposition can handle this case?

---

> ### Author Response · Authors · 2024-11-25
>
> We thank Reviewer 3Epu for his/her thoughtful feedback on our work.  We are glad to hear that the reviewer finds the paper “well-written”.  Regarding the reviewer’s comments and questions:
>
> **1-Technical contribution of our work**
>
> To explain our technical contribution, we note that without Theorem 1, the connection between the (unconditional) Vendi score and the Conditional-Vendi score (defined using the matrix-based conditional entropy) would remain unclear. This is because the conditional entropy $H(X|T)$, as defined in Sanchez Giraldo et al. (2014), does not correspond to the averaged entropy $H(X|T=t)$ over text instances $t \sim P_T$. It is important to note that the Vendi score relates only to instance-specific entropies $H(X|T=t)$.
>
> As explained in Item 1 of our general response, Theorem 1 in our paper addresses this gap. Specifically, under a mixture distribution on text $T$ with minimal assumptions, Theorem 1 provides an interpretation of the Conditional-Vendi score as the aggregation of Vendi scores across the different components of the mixture text distribution.
>
> **2- contextualization and interpretation of Theorem 1**
>
> We have included additional discussion in the revised introduction and section 5 to better explain the role of Theorem 1.
>
> **3- “showing some of the generations from the models in Figure 4”**
>
> We thank the reviewer for the suggestion. We have added the pictures generated by the models in Figure 4 of the main text to improve the clarity of the qualitative evaluation. The results can be found in Figure 25 in the revised Appendix B.7
>
> **4- “in light of Figure 6, it seems that it would be possible to achieve a high conditional Vendi score by a model that has poor alignment with the prompt.”**
>
> We would like to clarify that the video generation models in Figure 6 have generated videos that are aligned with the provided prompts. Due to space constraints, we included only 5 of the text prompts corresponding to the 16 generated videos displayed per cluster in Figure 6. As a result, 11 text prompts associated with the displayed videos were not shown in Figure 6, which may have led to the impression that the generation models produced poorly-aligned videos. We have clarified this point in the revised manuscript, and in the revised Appendix Figure 24, we have provided the complete set of prompts for the samples displayed in Figure 6.

---

> > ### Comment · Reviewer_3Epu · 2024-12-02
> >
> > Thanks to the authors for their response. I will keep my score.

---

> > > ### Author Response · Authors · 2024-12-03
> > >
> > > We thank Reviewer 3Epu for the time and feedback on our response.

---

### Author Response · Authors · 2024-11-25
**Authors' General Response**

We thank the reviewers for their thoughtful and detailed feedback on our work. In this general response, we would like to address the raised comments on our work’s technical contribution and experimental results.

**1- Significance of our technical contribution**

Regarding the comments on our work’s technical contribution, we would like to raise the following:

- To the best of our knowledge, our work is the first to study the diversity evaluation of conditional generative models by decomposing the diversity of their generated data into prompt-induced and model-induced terms.

- We note that neither the reference [1] (which proposes a formulation for matrix-based conditional entropy) nor the papers citing it have discussed the application of conditional entropy to generative models. To the best of our knowledge, the connection between the information measures in [1] and conditional generative models has not been explored in the literature.

- We believe the theoretical results in our work provide insights into the application of *matrix-based* conditional entropy for evaluating the model-induced diversity of conditional generative models. While Sanchez Giraldo et al. [1] introduce a notion termed “conditional entropy” $H(X|T)$, their defined $H(X|T)$ does not correspond to the expectation of prompt-conditioned entropies $H(X|T=t)$, which means
$$ H\bigl( X|T\bigr) \text{  } {\large\neq}  \text{  } \mathbb{E}_{t\sim P_T}\bigl[ H\bigl( X|T=t \bigr)\bigr] $$
Therefore, this **matrix-based** conditional entropy lacks the standard interpretation of Shannon conditional entropy, which is well-known to represent the average of instance-specific entropies, $H(X|T=t)$. Our work addresses this gap by establishing a connection between [1]’s definition of $H(X|T)$ and the averaged (unconditional) entropy over text prompts. Under *minimal assumptions*, Theorem 1 shows that this definition of $H(X|T)$ corresponds to an aggregation of the (unconditional) entropy $H(X|C=i)$ *within each component $C=i$ of a mixture prompt distribution $P_T$*.



**2- Golden metric for assessing the Conditional-Vendi score**

First, we would like to clarify that, to the best of our knowledge, the proposed Conditional-Vendi score is the first metric to evaluate the model-induced diversity of conditional generative models. In our numerical evaluation, we aimed to demonstrate the metric’s correlation with the ground-truth model-induced diversity of the models in scenarios where the ground-truth ranking of diversity is known across the tested generative models. This approach was necessary, because there are no existing metrics in the literature that could serve as a baseline for assessing the correlation with the Conditional-Vendi score.

Following Reviewer Uq1Z’s suggestion, in the revision, we have defined a baseline score as a “golden metric,” detailed in Appendix B.5 of the revised manuscript. This metric, which we refer to as the “GroundTruth-Cluster-Vendi,” assumes access to the underlying clusters of the generated samples and computes the average Vendi score within these sample clusters. As shown in Figure 22, the GroundTruth-Cluster-Vendi metric correlates well with the Conditional-Vendi score across the nine sets of numerical experiments conducted, where the ground-truth clusters were known (we provide a detailed explanation of the experimental setup in Item 3 below).

**3- Additional numerical support for the proposed diversity evaluation method**

In the revision, we have conducted 9 additional sets of numerical experiments to further evaluate the application of the Conditional-Vendi score in quantifying the model-induced diversity of conditional generative models. Specifically, we considered three SOTA text-to-image models: 1) Stable Diffusion XL, 2) Kandinsky 2.1, and 3) PixArt-$\Sigma$, along with three categories of text prompts: 1) Animals, 2) Fruits, and 3) Objects, resulting in a total of 9 different combinations.

For each of these 9 settings, we selected 10 different types within the category and generated images for 5000 prompts corresponding to these types. We then evaluated the Conditional-Vendi scores under two conditions: in the first, we used the original prompts; in the second, we replaced the specific category type (e.g., “apple”) with the general category name (e.g., “fruit”) in the prompt.

As shown in Figures 12–20, the Conditional-Vendi score increases consistently as more types within each category are introduced. Moreover, the score grows *significantly less rapidly* when the category type is explicitly specified in the prompt. These results indicate that the Conditional-Vendi score effectively mitigates the influence of prompt diversity, as desired, and primarily reflects the model-induced diversity of the generative model.

[1] Sanchez Giraldo et al., “Measures of entropy from data using infinitely divisible kernels”. IEEE Transactions on Information Theory, 2014

---

### Meta-Review · Area_Chair_WHgg · 2024-12-16

**Metareview:**

This work presents an information-theoretic approach to evaluate the diversity of prompt-based content generation. The proposed evaluation score defined in Section 4 consists of two parts: Conditional-Vendi and Information-Vendi. A statistical interpretation of the proposed score is presented in Section 5 (Theorem 1). The main criticism lies in the contribution. 3 out of the 4 reviewers think the proposed method is incremental at most and the proposed metric is derived from a simple transformation of existing ones. The reviewers also raise some questions on the numerical experiment design.

**Additional Comments On Reviewer Discussion:**

The authors respond by adding contribution clarifications and more experiments in the revision. The reviewers are not fully convinced and keep their original evaluations. I hope the reviewers’ comments provide useful feedback for the authors to improve the manuscript for future submission.

---

### Decision · Program_Chairs · 2025-01-22

Reject